# BARRA v1.0: The Bureau of Meteorology Atmospheric high-resolution Regional Reanalysis for Australia

Chun-Hsu Su[1], Nathan Eizenberg[1], Peter Steinle[1], Dörte Jakob[1], Paul Fox-Hughes[2], Christopher J. White[3,4], Susan Rennie[1], Charmaine Franklin[1], Imtiaz Dharssi[1], Hongyan Zhu[1]

[1] Bureau of Meteorology, Docklands, Victoria 3008, Australia
[2] Bureau of Meteorology, Hobart, Tasmania 7000, Australia
[3] Department of Civil and Environmental Engineering, University of Strathclyde, Glasgow, Scotland, UK
[4] Antarctic Climate and Ecosystems Cooperative Research Centre, Hobart, Australia

*Correspondence to*: C.-H. Su (chunhsu.su@bom.gov.au)

**Abstract.** The Bureau of Meteorology Atmospheric high-resolution Regional Reanalysis for Australia (BARRA) is the first atmospheric regional reanalysis over a large region covering Australia, New Zealand and southeast Asia. The production of the reanalysis with approximately 12 km horizontal resolution – BARRA-R – is well underway with completion expected in 2019. This paper describes the numerical weather forecast model, the data assimilation methods, and the forcing and observational data used to produce BARRA-R, and analyses results from the 2003-2016 reanalysis. BARRA-R provides a realistic depiction of the meteorology at and near the surface over land as diagnosed by temperature, wind speed, surface pressure, and precipitation. Comparing against global reanalyses ERA-Interim and MERRA-2, BARRA-R scores lower root-mean-square errors when evaluated against (point-scale) 2 m temperature, 10 m wind speed and surface pressure observations. It also shows reduced biases in daily 2 m temperature maximum and minimum at 5 km resolution, and a higher frequency of very heavy precipitation days at 5 and 25 km resolution when compared to gridded satellite and gauge analyses. Some issues with BARRA-R are also identified: biases in 10 m wind, lower precipitation than observed over the tropical oceans, higher precipitation over regions with higher elevations in south Asia and New Zealand. Some of these issues could be improved through dynamical downscaling of BARRA-R fields using convective-scale (< 2 km) models.

## 1 Introduction

Reanalyses are widely used for climate monitoring and studying climate change as they provide long-term spatially complete records of the atmosphere. This is achieved by using data assimilation techniques that produce an observation-constrained model estimate of the atmosphere. They draw short-term model states towards observations from multiple, disparate sources to form an atmospheric analysis. A physically realistic model provides the means to infer atmospheric states at locations without observations from the limited collection of irregularly distributed observations.

Global-scale reanalyses using global atmospheric circulation models (GCMs) have advanced in quality and quantity during the past two decades (Dee et al., 2014; Hartmann et al., 2013). At present, the available global reanalyses established for the satellite era include the NCEP/NCAR reanalysis at 210 km horizontal resolution (Kalnay et al., 1996), the Japanese 55-year Reanalysis (JRA-55) at 60 km (Ebita et al., 2011), the Modern-Era Retrospective analysis for Research and Applications-2 (MERRA-2) at about 50 km (Gelaro et al., 2017) and the European Centre for Medium Range Weather Forecasts (ECMWF) ReAnalysis Interim (ERA-Interim) at ~79 km (Dee et al., 2011). The latter is currently being replaced by the new ERA5 ~31 km reanalysis (Hersbach and Dee, 2016). These global reanalyses have the advantage of providing globally consistent information, but at the expense of spatial resolution. With resolutions typically greater than 50 km, they may be deficient in accounting for important subgrid variations in meteorology over heterogeneous terrains and islands, and across irregular coastlines, and other small-scale processes (Mesinger et al., 2006; Randall et al., 2007, and references therein).

To address these shortcomings, the development in global reanalysis has also driven concurrent efforts in statistical approaches and dynamical downscaling (e.g., Dickinson et al., 1989; Fowler et al., 2007; Evans and McCabe, 2013). The latter typically embeds a high-resolution meteorological  model within a global reanalysis, where effects of small-scale forcing and processes such as convection are modelled. Such development is supported by improvements in non-hydrostatic models that run at high resolution in operational numerical weather prediction (NWP) (e.g., Clark et al., 2016). Regional reanalyses are emerging as a step further in this direction. The first regional reanalysis was the North America Regional Reanalysis (NARR, Mesinger et al., 2006). More recent examples include the Arctic System Reanalysis (ASR, Bromwich et al., 2018), Indian Monsoon Data Assimilation and Analysis (IMDAA, Mahood et al., 2018) and Uncertainties in Ensembles of Regional Reanalyses (UERRA) in Europe (Borsche et al. (2015) and therein). In contrast to dynamically downscaled global reanalyses, observations are used in regional reanalyses in the same way as in the global ones to reduce model errors in high-resolution simulations (Bollmeyer et al., 2015). The resulting reanalyses are expected to have better representations of frequency distributions, extremes and actual space and time-dependent variability (particularly for near-ground variables). UERRA consists of four regional reanalyses  developed by the Swedish Meteorological and Hydrological Institute (SMHI), Météo France, Deutscher Wetterdienst (DWD), and UK Met Office (UKMO), producing an ensemble of high resolution (5–25 km)  regional reanalyses of essential climate variables. The SMHI's HARMONIE [Hi-Resolution Limited Area Model (HIRLAM) Aire Limitée Adaptation Dynamique Développement International (ALADIN) Regional/Mesoscale Operational NWP in Europe] reanalysis has entered production for the Copernicus Climate Change Service (Ridal et al., 2017).

Regional reanalyses provide significant added value to their global counterparts in diverse applications ranging from traditional climate studies to industry applications, including regional climate change assessments that include local impact studies (e.g., Fall et al., 2010) and extreme events reconstruction (e.g., Zick and Matyas, 2015). As the regional reanalyses are generally produced with high spatial as well as temporal resolution, the extremes of variables at local scales may be quantified more accurately. They are also an alternative reference to evaluate climate projections (e.g., Ruiz-Barradas and Nigam, 2006; Radic

and Clarke, 2011). At the same time, embedded forecast models can be used within the framework of the Coordinated Regional Climate Downscaling Experiment (CORDEX, Martynov et al., 2013) within a seamless framework for weather and climate prediction, where model deficiencies that differ in spatial and time scales can be more readily understood (Brown et al., 2012). They also offer useful data sets for designing new infrastructure, particularly if they are sufficiently long and spatially relevant

to define the likelihood of extremes. For renewable energy production, they can provide valuable information on intermittency (e.g., wind lull) and covariability (e.g., correlation spatially or between variables) of phenomena. For instance, COSMO (Consortium for Small-scale Modelling) 6 km reanalysis has shown the potential to provide realistic sub-daily representations of winds at 10 to 40 m height (Borsche et al., 2016), and to resolve small-scale cloud structures (Bollmeyer et al., 2015). NARR was used to define a climatology of surface wind extremes (Malloy et al., 2015), and 30-year trends in wind at hub height

(Holt and Wang, 2012) over northern America.

To date, while regional reanalyses exist for North America, Europe and India, no atmospheric regional reanalysis for the Australasian region has been produced. To close this gap, the Bureau of Meteorology Atmospheric high-resolution Regional Reanalysis for Australia (BARRA, Jakob et al., 2017) has been produced. BARRA is the first atmospheric regional reanalysis that covers Australia, New Zealand, southeast Asia, and south to the Antarctic ice edge (Figure 1). It is produced by the

Australian Bureau of Meteorology (Bureau), with sponsorship from state fire and governmental agencies across Australia, because of the important advantages it provides for planning and management to reduce risks due to extreme weather events including bushfires. For instance, BARRA will address the lack of accurate climate information on highly variable surface winds over large areas of Australia due to the low density of the surface observation network in remote areas. BARRA covers a 29-year period from 1990 to 2018, with possible further extensions back and forward in time. The BARRA project delivers

a whole-of-domain reanalysis (identified as BARRA-R) with approximately 12 km horizontal resolution, and additional convective-scale (1.5 km horizontal grid-length) downscaling (BARRA-x), nested within BARRA-R, centred on major Australian cities to generate additional high-resolution information needed for local-scale applications and studies. These resulting gridded (12 km and 1.5 km) products include a variety of 10 min to hourly surface parameters, describing weather and land-surface conditions, and hourly upper-air parameters covering the troposphere and stratosphere. The fields on standard

pressure levels are generated from vertical interpolation of model-level fields. BARRA serves to lay the foundation for future generations of reanalyses at the Bureau and to further develop its capabilities to produce seamless climate information that integrates its observational networks and NWP programme.

In this paper, we describe the forecast model, data assimilation methods, and the forcing and observational data used to produce BARRA-R in Section 2. Section 3 provides an initial assessment of the reanalysis system over the first 14 years 2003-2016,

with a focus on analysing the quality at or near the surface; Section 4 concludes with a brief summary of our findings.

## 2 The BARRA-R reanalysis

The development of BARRA builds on the Bureau's experience in operational (deterministic) NWP forecasting over the Australian region using the Australian Community Climate and Earth-System Simulator (ACCESS)-R system (Bureau of Meteorology, 2010; 2013; Puri et al., 2013), and BARRA-R is produced using the UKMO's system in UERRA (based on Jermey and Renshaw, 2016) but without the ensemble component. An ensemble NWP forecast system is currently under development at the Bureau. BARRA-R is produced by running a limited-area meteorological forecast model forced with global reanalysis boundary conditions, drawn closer to observations via data assimilation. This section provides an overview of these components while more technical details are included in the references.

### 2.1 Forecast model

The Unified Model (UM, Davies et al., 2005) is the grid-point atmospheric model used in BARRA-R and ACCESS. It uses a non-hydrostatic, fully compressible, deep-atmosphere formulation and its dynamical core (Even Newer Dynamics for General atmospheric modelling of the environment, ENDGame) solves the equations of motion using mass-conserving semi-implicit, semi-Lagrangian time-integration methods (Wood et al., 2014). The model includes a comprehensive set of parametrizations, including a modified boundary layer scheme based on Lock et al. (2000), a variant of Wilson and Ballard (1999) for mixed-phase cloud microphysics, the mass flux convection scheme of Gregory and Rowntree (1990), and the radiation scheme of Edwards and Slingo (1996), which have all since been improved. Other parametrized sub-grid scale processes include , fractional cloud cover  and orographic drag. More details on all of the physics schemes can be found in Walters et al. (2017a).

The prognostic variables are three-dimensional wind components, virtual dry potential temperature and Exner pressure, dry density, and mixing ratios of moist quantities. The model is discretized on a horizontally staggered Arakawa C-grid (Arakawa and Lamb, 1977) and a vertically staggered Charney-Phillips grid (Charney and Phillips, 1953). The staggered arrangement of grid points allows for accurate finite differencing but results in different model fields located on staggered grids displaced by half a grid spacing along both axes. Data has been left on the staggered grids to allow users to apply the most appropriate re-gridding methods suited for given applications. The vertical levels smoothly transition from terrain-following coordinates near the surface to constant height surfaces in the upper atmosphere (Davies et al., 2005).

BARRA-R uses version 10.2 of the UM and is configured with 70 vertical levels extending from near the surface to 80 km above sea level: 50 model levels below 18 km, and 20 levels above this. While configured with this height based on ACCESS-R, we have more confidence in the data up to a height of 25-30 km where we have most information from observations. The horizontal domain of BARRA-R spans from 65.0° to 196.9° E, -65.0° to 19.4° N (Figure 1), with constant latitude and longitude increments of $0.11° \times 0.11°$ and $1200 \times 768$ grid points in the horizontal. Our choice of the horizontal resolution follows the deterministic component of the UKMO reanalysis and the IMDAA reanalyses. The model was run to produce 12-

hour (12h) forecasts in each 6-hourly cycle (see Section 2.2) to give extra data for driving dynamical downscaling within the domain.

The model parametrizations in BARRA-R are inherited from the UKMO Global Atmosphere (GA) 6.0 configurations described in Walters et al. (2017a). The GA6 configurations are also suited for limited-area models with resolutions > 10 km, but with some modifications:

i.      A variable Charnock coefficient is used in surface heat exchange over the sea to improve the tropical Pacific air-sea exchange (Ma et al., 2015).

ii.      The heat capacity of "inland water canopy" is set to $2.11 \times 10^7$ J K$^{-1}$m$^{-2}$ for modelling lakes. This improves the diurnal cycle over inland waters. By contrast, grid cells containing salt lakes in Australia are modelled as bare soil surface (for Lake Eyre and Lake Frome) and vegetated surface (e.g., Lake Lefroy, Lake Ballard).

iii.      For its deep convective mass flux scheme, a grid-box dependent convective available potential energy (CAPE) closure scheme is chosen to limit the role of parameterized convection. When vertical velocity exceeds the given threshold of 1 m/s, the vertical velocity dependent CAPE closure is chosen to release the convective instability efficiently (Zhu and Dietachmayer, 2015). These changes aim to improve the model stability.

iv.      The river routing scheme has been turned off because it is not designed for a limited-area model. Therefore, there is no routing of runoff from inland grid points out to sea and inland water bodies, and soil moisture is not affected by this hydrological process.

The characteristics of the lower boundary, climatological fields and natural and anthropogenic emissions are specified using static ancillary fields. These are created as per Walters et al. (2017a, Table 1), with the exceptions of the land-sea mask and canopy tree heights. The land-sea mask is created from the 1 km resolution International Geosphere–Biosphere Programme (IGBP) land cover data (Loveland et al., 2000), and the canopy tree heights are derived from satellite light detection and ranging (LiDAR, Simard et al., 2011; Dharssi et al., 2015). Climatological aerosol fields (ammonium sulphate, mineral dust, sea salt, biomass burning, fossil-fuel black carbon, fossil-fuel organic carbon, and secondary organic (biogenic) aerosols) are used to derive the cloud droplet number concentration. Absorption and scattering by the aerosols are included in both the shortwave and longwave.

**2.1.1 Land surface**

The UM uses a community land surface model, the Joint UK Land Environment Simulator (JULES, Best et al., 2011). It models partitioning of rainfall into canopy interception, surface runoff and infiltration, and uses the Richards' equation and Darcy's law to model soil hydrology. Sub-grid scale heterogeneity of soil moisture is represented by the Probability Distributed Moisture (PDM) model (Moore, 2007). A nine-tile approach is used to represent sub-grid scale heterogeneity in land cover,

with the surface of each land point subdivided into five vegetation types (broadleaf tree, needle-leaved trees, temperate C3 grass, tropical C4 grass and shrubs) and four non-vegetated surface types (urban, inland water, bare soil and land ice). It describes a 3 m soil column with a 4-layer soil scheme with soil thicknesses of 0.1, 0.25, 0.65 and 2.0 m, and models vertical heat and water transfer within the column with van Genuchten hydraulic parameters. The JULES urban parameters are
optimised for Australia as described by Dharssi et al. (2015).

### 2.1.2 Soil moisture

For the 1990-2014 period, soil moisture fields in BARRA-R are initialised daily at 06 UTC using soil moisture analyses from an offline simulation of JULES, at 60 km resolution, driven by bias corrected ERA-Interim atmosphere forcing data, using methods described in Dharssi and Vinodkumar (2017) and Zhao et al. (2017). The simulation used a 10-year long spin-up
period and then was run continuously for the 1990 to 2014 period. The near-surface soil moisture analyses are found to have good skill for the Australian region when validated against ground-based soil moisture observations (Dharssi and Vinodkumar, 2017). As the offline runs were terminated at the end of December 2014, the daily initialization scheme is continued with soil moisture analyses from the Bureau's global NWP system – ACCESS-G (Bureau of Meteorology, 2016). These external soil moisture analyses are downscaled to the BARRA-R grid using a simple method that takes into account differences in soil
texture. As well, in each 6-hourly cycle, a land surface analysis is conducted within BARRA (see Section 2.2). The daily initialisation was conducted with the purpose of avoiding spurious drift in the BARRA moisture fields and reducing the time needed to spin up from ERA-Interim initial conditions. However, as multiple parallel production streams are needed to produce the reanalysis (see Section 2.2), a discontinuity in soil moisture in the bottom two layers exists between successive production streams, although soil moisture in the top two layers becomes stable after one-month of runs. A discontinuity occurring at the
2014-2015 changeover has recently been reported by BARRA data users. These impacts, particularly on forested regions where trees extract water from the deep soil layers, are under investigation.

### 2.1.3 Boundary conditions

The BARRA-R sequential data assimilation process is initialized using ERA-Interim analysis fields (see Section 2.2), after which the only relationship with ERA-Interim is solely through the lateral boundary conditions. Hourly lateral boundary
conditions for BARRA-R are interpolated from ERA-Interim's 6-hourly analysis fields at $0.75° \times 0.75°$ resolution. The rim width of the boundary frame is 0.88°.

The land boundary is provided by a land surface analysis (Section 2.2). Daily sea-surface temperature (SST) and sea ice (SIC) analysis at $0.05° \times 0.05°$ resolution from reprocessed (1985-2007, Roberts-Jones et al., 2012) and near real-time (NRT) Operational Sea Surface Temperature and Ice Analysis (OSTIA, Donlon et al., 2012) are used as lower boundaries over water
after being interpolated to the UM grid. The NRT data is used from January 2007. OSTIA is widely used by NWP centres and

operational ocean forecasting systems, owing to their short real-time latency. Even though the re-processed and NRT data do not constitute a homogeneous timeseries, OSTIA is favoured over other SST reanalyses owing to its higher spatial resolution. Masunaga et al. (2015, 2018) have shown steep SST gradients, unresolved by coarse SST reanalyses, can influence the organization of long-lived rain bands and enhancement or reduction of surface convergence, and this is particularly problematic for atmosphere-only reanalyses as thermal structure and motions in the marine atmospheric boundary layer are not well constrained by data assimilation.

## 2.2 Data assimilation system

The BARRA-R analysis scheme is based on fixed deterministic atmospheric and land surface assimilation systems used by the UKMO in UERRA (Jermey and Renshaw, 2016) and IMDAA (Mahmood et al., 2018). BARRA-R uses a sequential data assimilation scheme, advancing forward in time using 6-hourly analysis cycles centred at synoptic hours $t_0 = 0$, 6, 12 and 18 UTC, and 12h forecast cycles from $t_0$-3h (Figure 2).

In each analysis cycle, available observations, distributed across a 6h analysis window $t_0$-3h $\leq$ t < $t_0$+3h, are combined with the prior information of the model forecast from the previous cycle (the background state), to provide a more accurate estimate of the atmosphere over this window. This first involves a 4-dimensional variational (4DVar) analysis of the basic upper-air atmospheric fields (wind, temperature, specific humidity, pressure) with conventional and satellite observations (see below). 4DVar is favoured over 3DVar as it takes account of time tendency information in the observations and this has a positive impact on the resulting forecasts (Rawlins et al., 2007). The UKMO's VAR assimilation system (version 2016.03.0) is used. The 4DVar uses a linear perturbation forecast (PF) model (Lorenc 2003; Rawlins et al., 2007, Lorenc and Payne, 2007), which uses a simpler model state linearised about a 'guess' trajectory (i.e., tangent linear model) with a lower resolution (0.33° cf. 0.11°) than the full forecast model. The lower resolution is chosen to limit computational costs. The PF model uses a simplified set of physical parameterizations including a simple boundary layer, cloud latent heat release, large-scale precipitation and convection. In other words, it is assumed that the lower-resolution corrections to the background state (i.e. increments), interpolated to a higher resolution, are suitable corrections for the full model. The analysis increments from 4DVar valid at $t_0$-3h are added to the background state at $t_0$-3h to produce an improved initial condition for the forecast model to perform the next 12h forecast from $t_0$-3h to $t_0$+9h. A constraint of zero analysis increments is specified at the model boundary such that BARRA-R relies on the driving model ERA-Interim to define large-scale flow and other atmospheric conditions (Section 2.1.3). The observation departure statistics of the analysis, which are differences between the analysis and observations, are shown to be less than those of the model background (Supplementary Material, Table S1). The assimilation is therefore behaving as desired by drawing the model towards observations for nearly all observational types.

The variational method minimises a cost function whose two principal terms penalise distance to the background state and distance to the observations. The two terms are squared differences weighted by the inverse of their corresponding error

covariances. In BARRA-R, the background error covariance has been estimated by a smooth parameterised approximation to climatology tuned by forecast differences (Ingleby, 2001). Accordingly, the estimated background error covariance is invariant between successive analysis windows, but is time varying within the analysis window. The cost function also includes a pressure-based energy norm that serves as a weak constraint digital filter to suppress spurious fast oscillations associated with gravity-inertia waves produced in model forecasts when analysis increments are added to the background state (Gauthier and Thépaut, 2001).

The initial land surface state can have a significant impact on short-term forecasts of screen-level temperature and humidity, and its quality can also be improved through data assimilation. An Extended Kalman Filter (EKF) using observations of 2 m temperature and humidity is used to analyse the BARRA land state at every 6 hour cycle and provide analyses of soil moisture, soil temperature and skin temperature as described by Dharssi et al. (2012). The assimilation of satellite-retrieved soil moisture is not attempted here as it has not been realised in ACCESS. The UKMO's SURF analysis system (version 2016.07.0) is used to perform EKF. The Jacobian, which relates observed variables to model variables, for the Kalman gain matrix is estimated using finite difference by perturbing each model variable to be analysed in 40 perturbations and performing short 3-hour forecasts. Here JULES (version 3.0) is run in the standalone mode, decoupled from the UM. The BARRA-R land state is reconfigured with EKF-derived surface analyses at every $t_0$.

Note that the last 6h forecast of a model run represents the prior state estimates needed for the next analysis cycle. The forecast fields valid at $t_0$-3h, $t_0$-2h and $t_0$-1h are discarded, as these fields may still be influenced by transient artefacts due to the slight imbalance introduced by the addition of the analysis increments. It is already noted that this effect is also mitigated with the energy norm in the 4DVar's cost function that penalises the unbalanced structure in the increments.

The reanalysis is produced with multiple parallel production streams to speed up production. Each stream has a month of spin-up time from the ERA-Interim initial conditions before production data is archived, with most streams producing one year of reanalyses. Trials have shown that a one-month period is sufficient spin-up for the atmosphere (Renshaw et al., 2013) and top levels of soil moisture, but insufficient for soil moisture in the deeper layers.

## 2.3 Observations

Conventional observations from land surface stations, ships, drifting buoys, aircrafts, radiosondes, wind profilers, and satellite observations, namely retrieved wind, radiances and bending angle, are assimilated in BARRA-R. The various observational types are chosen as they have been assimilated in the Bureau's operational NWP systems; other observational types, such as clear-sky radiances, have not been assimilated due to resource constraints. Rain observations from radar and gauges are also not assimilated as their assimilation schemes are still being tested for operational NWP. As listed in Table 1, the data sets are pragmatically taken from multiple sources, as they are being prepared during the production runs. Most of the observations

prior to 2009 are supplied by ECMWF, and the satellite radiance data from 2017 and onwards are extracted from the UKMO operational archive.

The Bureau's archived observational data is also used to support this work, especially for the cycles from 2010 onwards. BARRA-R also assimilates additional high frequency (10 min) land surface observations from automatic weather stations in
Australia, and locally derived satellite atmospheric motion vectors (AMV). Ground positioning system (GPS) radio occultation bending angle data up to 2009 is provided by the Radio Occultation Meteorology Satellite Application Facility (ROM SAF). Additional land surface observations over New Zealand are extracted from their National Climate Database (CliFlo, 2017). The 4DVar assimilation of local AMV (Le Marshall et al., 2013) and GPSRO (Le Marshall et al., 2010) has been shown to improve operational forecasts.

Before being assimilated, observations are screened to select the best quality observations, remove duplicates and reduce data redundancy via thinning, using the UKMO's Observing Processing System (OPS) (based on version 2016.03.0) (Rawlins et al., 2007). There are per-cycle quality controls performed based on the method of Lorenc and Hammon (1988). Observations significantly different from the model background are rejected when exceeding a threshold calculated by a Bayesian scheme, unless they are consistent with other observations nearby. The observational error variances and thinning distances are
established at the UKMO and the Bureau for their NWP systems. For the surface, sonde and aircraft observations, an observation automatic monitoring system performs monthly blacklisting of sites that show consistently large differences with BARRA-R's forecast over a one-month period. The system also calculates bias corrections for surface pressure and for aircraft and sonde temperature.

For the satellite data, instruments and their individual channels are rejected when they become unreliable. The blacklisting is
informed by the work of the ECMWF and MERRA-2 reanalysis teams. Further, airmass-dependent variational bias correction is applied to satellite radiances as part of the assimilation process, allowing the time-varying corrections to fit drifts in instrumental bias (Harris and Kelly, 2001; Dee and Uppala, 2008). The bias corrections were calculated monthly, with the satellite radiances during the first month of each production stream not assimilated. There are abrupt changes to the amount of satellite data assimilated at the start and end of satellite missions and the various observational data archives. In some cases,
changes occur when corrections were made to the observation screening and thinning rules mid-production of the 2010-2015 reanalyses. The impacts of such changes, known to cause artificial shifts and spurious trends in a reanalysis (e.g., Thorne and Vose, 2010; Dee et al., 2011) are still to be investigated for BARRA-R.

**3 Preliminary evaluation**

Our evaluation focuses on three areas: surface variables, pressure-level temperature and wind, and precipitation. For the surface
variables, we compare BARRA-R against point-scale observations and gridded analyses of observations for 2 m temperature.

For the pressure levels, we evaluate BARRA-R against point-scale observations of temperature and wind, and examine the timeseries of the bias between BARRA-R and the global reanalyses. Finally, as rain observations are not assimilated in BARRA-R, gridded analyses of rain observations from gauges and satellites are used to provide the best independent reference in this study.

## 3.1 Surface

### 3.1.1 Point-scale evaluation of 2 m temperature, 10 m wind speed and surface pressure

The $t_0+6h$ model forecasts of 2 m (screen) temperature, 10 m wind speed and surface pressure are evaluated against land observations. These observations have only an indirect relation to the forecasts as they are not used in the analysis for the associated cycle $t_0$. Since errors tend to grow with the forecast range, the assessment places an upper bound on the true errors of the analysis fields between time $t_0$ and $t_0+3h$. These fields are interpolated from the model levels using surface similarity theory (Walters et al., 2017a). The ERA-Interim $t_0+6h$ forecasts from 0 and 12 UTC and the MERRA-2 hourly time-averaged forecast fields (M2T1NXSLV) are also evaluated to serve as benchmarks. It is not ideal to directly compare reanalyses with different resolutions, and interpolating them onto common (observed) locations diminishes some of the improvement achieved by BARRA-R relative to coarser reanalyses. Nonetheless, we undertake the latter to assess whether the models contain finer-scale information captured by point measurements; it therefore does not provide an assessment of the true quality of the reanalyses at their native resolutions.

To correct representativity errors in both reanalyses, their model values at (modelled) land grid cells are interpolated to the observation times and the station locations via bilinear interpolation in time and in the horizontal direction. Height corrections are applied to the interpolated fields to match the station heights: the corrections to the screen temperature is based on dry adiabatic lapse rate (Sheridan et al., 2010), 10 m wind speed is based on Howard and Clark (2007), and the correction to surface pressure is based on the hydrostatic equation under a constant lapse rate. As the observations are irregularly distributed in time, we consider all observations within a $t_0+5h$ to $t_0+7h$ time window, with $t_0$ being 0 and 12 UTC, and the model grids are linearly interpolated to the observation times. Root-mean-squared difference (RMSD), Pearson's linear correlation, additive bias and variance bias are calculated at each station, with $bias = mean(d_m) - mean(d_o)$, the variance bias as $Mbias = var(d_m)/var(d_o) - 1$ to capture differences in the dispersion, where $var(*)$ computes the variance in time.

Boxplots in Figure 3 show the distribution of scores across 900-1500 stations in the BARRA-R domain. BARRA-R shows better agreement with the point observations than the global reanalyses for all three surface variables by most of the measures. This result is expected since BARRA-R resolves near-surface features below 50 km horizontal scale, and assimilates more surface observations over Australia and New Zealand. In particular, BARRA-R shows lower RMSD at about 80% of the stations for screen temperature and 10 m wind speed, and at 70% of stations for surface pressure (see Figure S1 of the

Supplementary Material). At closer inspection in Figure 4(a), a percentile comparison plot of screen temperature deviation from monthly mean indicates that the frequency distribution of BARRA-R temperature is closer to that of the observations than ERA-Interim, particularly in regimes below 25% percentiles and above 90% percentiles.

For 10 m wind speed, negative biases for variance exist in all the reanalyses assessed in this paper. Figure 4(b) shows that 10 m wind speeds are positively biased during light wind conditions and vice versa during strong wind speeds. There are many possible reasons for under-estimating strong winds: the inaccurate descriptions of boundary layer mixing and form drag for sub-grid orography, and of surface properties such as land cover and vegetation types. Changing the fractional area of the vegetation canopy modifies scalar roughness of the vegetated tiles, affecting the wind speed. The seemingly linear variation in wind speed is known in the global reanalyses (e.g., Carvalho et al., 2014), and Rose and Apt (2016) attributed the problem of wind underestimation to inaccuracy in modelling wind speeds in unstable atmospheric conditions.

Pressure is a large-scale variable which is likely to be better represented by a global model than a limited-area model. However, the BARRA-R estimates of point-scale surface pressure are more accurate in topographically complex regions and coastlines (see Figure S1 of the Supplementary Material), where the estimates from the coarser reanalyses are less representative.

### 3.1.2 Comparison with gridded analysis of observed 2 m temperature

The reanalyses are compared against a gridded daily $0.05° \times 0.05°$ analysis of station maximum and minimum 2 m temperature data from the Australian Water Availability Project (AWAP, Jones et al., 2007). The AWAP grids are generated using an optimised Barnes successive-correction method that applies weighted averaging of the station data. Topographical information is included by using anomalies from long-term (monthly) averages in the analysis process. The AWAP analysis errors for maximum temperature are larger near the coast around northwest Australia and around the Nullarbor Plain, due to strong temperature gradients between the coast and inland deserts and a relatively sparse network (Jones et al., 2007). The coast of Western Australia and parts of Northern Territory are likely to share this similar analysis issue. The analysis errors are larger for minimum temperature, especially over Western Australia and the Nullarbor Plain.

Figure 5 shows the differences for 2007-2016 averages in daily maximum and minimum temperature from AWAP, ERA-Interim, MERRA-2 and BARRA-R. The daily statistics are derived from 3-hourly forecast fields of ERA-Interim and hourly fields of MERRA-2 and BARRA-R. While inherent biases due to sampling are expected, this comparison also highlights the advantage of higher frequency data when examining lower and upper tail statistics. BARRA-R shows cold and warm biases (relative to AWAP) of around 1 K in daily maximum and minimum temperature respectively, particularly over the eastern region. MERRA-2 also shows similar levels of biases but with different signs and variability. BARRA-R and MERRA-2 agree better with AWAP than ERA-Interim, which reports differences (in mean) up to 5 K in magnitude. The reduced amplitude of the diurnal cycle of temperature is a long-standing problem in the UM; experiments have shown that changes to the

representation of the land surface (e.g., reductions in the amount of bare soil and changes to scalar roughness and albedo of vegetated tiles) reduce clear-sky biases (Bush et al., 2019).

Figure 6 shows the monthly means of the differences in daily maximum and minimum temperature between the reanalyses and AWAP averaged across Australia. Here the OSTIA SST anomaly timeseries is also included, and it does not show a visible discontinuity at 2006/2007 (Section 2.1.3). The maximum temperature in BARRA-R appears cooler than AWAP after a strong La Nina event in 2010-2011, while the global reanalyses also show cooler trends in biases after 2010. BARRA-R and ERA-Interim show smaller levels of temporal variability than MERRA-2. The minimum temperature in BARRA-R does not show an obvious trend but is warmer during 2010-2011 when ERA-Interim and MERRA-2 are cooler. These changes do not coincide with the change in soil moisture initialization in 2014-2015 (Section 2.1.2) or OSTIA SST.

## 3.2 Pressure levels

To assess BARRA-R in the atmosphere, we compare the $t_0$+6h forecasts on pressure levels with radiosonde and pilot wind observations at 0 and 12 UTC on standard pressure levels ranging from 1000 to 10 hPa, using the harmonized data set produced by Ramella Pralungo et al. (2014a; 2014b). The pressure-level fields of BARRA-R and ERA-Interim's analyses at time $t_0$ are also compared, even though they are not independent from the observations; such comparisons only provide baselines to interpret the relative quality of the BARRA-R forecasts. Similar comparisons with ERA-Interim's twice-daily forecasts at these observation times are also not possible because they start from 0 and 12 UTC. The model data is interpolated horizontally to the sonde and pilot launch locations via bilinear interpolation, and RMSD is calculated at each location and pressure level. The resulting boxplots of RMSD are shown in Figure 7. Depending on the pressure level and parameter evaluated, between 54 to 203 sites were available. There is a marked variability in RMSD with the pressure levels, particularly for wind speed, due to reasons such as variations in the number of observing sites, increasing sonde drift error on ascent, and differences in dynamic range of the fields with height. A markedly higher RMSD for wind speed occurs at 200 hPa, a height a which the jet stream can be located.

It is difficult to discern the differences between the two analyses, suggesting that they perform similarly from assimilating the same observations. Assimilation at a coarser resolution of 0.33° (cf. 0.11° of the forecast model) in BARRA-R does not drastically improve 0.75° representations of temperature and wind at these pressure levels and at point scales. There are also small differences between the analyses and BARRA-R background, indicating that the 0.11° forecast model does not degrade from the lower-resolution analysis of BARRA-R but also does not improve upon the ERA-Interim's 0.75° representation of these fields at the observation locations.

Figure 8 compares BARRA-R's 0 UTC analysis of air temperature at 850, 700 and 500 hPa against the analyses from ERA-Interim and MERRA-2 (M2I3NPASM). BARRA-R is cooler at 500 hPa across the domain, and warmer at 850 hPa in the

tropics than the global reanalyses, and the monthly differences in the zonal mean are of the order of 1 K. BARRA-R also shows a cooling shift at 700 and 500 hPa in the tropics, and a warming shift south of 40°S after 2010. But when compared against MERRA-2, in the tropics, BARRA-R is warmer at 700 hPa, and the apparent shift in BARRA-R is also seen in MERRA-2 (relative to ERA-Interim) at these levels.

## 3.3 Precipitation

We consider three reference gridded data sets to compare with the reanalyses. First is the 0.05° × 0.05° rain gauge analysis of daily accumulation over Australia from AWAP, produced using the Barnes method where the ratio of observed rainfall to monthly average is used in the analysis process (Jones et al., 2009). There is a north-south gradient in the AWAP analysis errors with larger analysis errors in the northern tropical regions, where length scales of convective rainfall events are shorter and more variable (Jones et al., 2009). Second is the 1° × 1° (full data daily) rain gauge analysis over the domain from the Global Precipitation Climatology Centre (GPCC version 2018, Ziese et al., 2018), created using an empirical weighting-based interpolation method described in Becker et al. (2013). As with AWAP, GPCC is less accurate in regions where station scarcity and high precipitation variability coexist. For instance, different GPCC interpolation methods can yield very different analyses over the south Asia region (Becker et al., 2013). The third reference is the 0.25° × 0.25° satellite-based analysis of 3-hourly rain rates from the Tropical Rainfall Measuring Mission (TRMM) multi-satellite precipitation analysis (TMPA 3B42 version 7, Huffman et al., 2006). TMPA 3B42 combines precipitation estimates from various satellite systems and rain gauge monthly analysis. Satellite-derived estimates of convective precipitation are largely accurate in the low latitudes (Ebert et al., 2007, Chen et al., 2013), but the TMPA product is less accurate over the ocean due to the absence of local observations used for gauge adjustments (Sapiano and Arkin, 2009), and south of 40°S due to limited local cross-sensor calibration (Huffman et al., 2008). TRMM often underestimates precipitation in high-latitude regions with significant topography due to difficulties of satellite retrievals over snow covered surfaces and/or due to the high elevations (Barros et al. 2006; Matthews et al. 2013). TRMM is also known to underestimate light rainfall and drizzle over subtropical and high-latitude oceans (Berg et al., 2010). In addition to these considerations, there are inherent limitations in comparing the reanalyses with AWAP, GPCC and TMPA. Specifically, products with coarser grids tend to over-represent low-threshold events occurring at spatial scales smaller than their grid sizes and under-represent high-threshold events. Further evaluation of BARRA-R precipitation estimates against point gauge observations and AWAP are reported in Acharya et al. (2019).

Neither BARRA-R nor ERA-Interim assimilated rainfall observations. Precipitation estimation from their forecast models is constrained by other observation types. Following Section 2.1, in BARRA-R, the microphysics scheme based on Wilson and Ballard (1999) parameterises the atmospheric processes that transfer water between the four modelled states of water (vapour, liquid droplets, ice, and raindrops) to remove moisture resolved on the grid scale. As the 12 km model is not "storm resolving", BARRA-R uses the mass flux convective parameterization scheme of Gregory and Rowntree (1990) with the CAPE closure

to model sub-grid scale precipitating and non-precipitating convection using an ensemble of cumulus clouds as a single entraining-detraining plume. Such a scheme prevents unstable growth of cloudy structures on the grid, which is otherwise required for explicit vertical circulations to develop (Clark et al., 2016). The modelled convection also works independently at each grid point, and the model can only predict the area-average rainfall, instead of the spectrum of rainfall rates. Consequently, BARRA-R's precipitation estimates from sub-grid convection will be more erroneous than those for large-scale precipitation. In other words, the accuracy of BARRA-R is expected to worsen during the warm season and at low latitudes, and to improve during cooler season and at high latitudes where non-convective precipitation is dominant. To allow the UM to spin-up from the analysis increments, we examine the quality of the precipitation accumulation between $t_0$+3h to $t_0$+9h, by comparing against gridded data sets. This also addresses the issue that the UM yields excess precipitation at analysis time ($t_0$-3h) due to a temporary imbalance in the moisture fields, by allowing time for the model to adjust and remove the excess. For ERA-Interim, we used its first 12h accumulation, which is considered the most accurate (Kallberg, 2011).

### 3.3.1. Mean annual precipitation and frequency of rain days

Figure 9, row (i) compares the ten-year (2007-2016) annual mean precipitation estimated from the five data sets. A close-up over Australia can be found in Figure S2 of the Supplementary Material. BARRA-R provides a realistic depiction when compared with TMPA across the domain, but shows higher precipitation over the tropics and over the Tasman Sea and Southern Ocean. BARRA-R agrees very well with AWAP and GPCC over Australian land areas, reflecting the markedly higher precipitation in the northern tropics, and western Tasmania. It also agrees with GPCC over New Zealand. BARRA-R also shows better agreement with AWAP, GPCC and TMPA in some of the dry areas such as western Australia.

The frequency of days with three intensity regimes is examined next in Figure 9. First in row (ii), we examine the frequency of light rain days with amounts between [1,10) mm. The 1 mm threshold is chosen to account for the tendency of the model to create light "drizzle" events with very low rain rates. Even so, the two reanalyses show significantly more rain days in the tropics than TMPA and GPCC, and more rain days than TMPA over the Southern Ocean. TRMM is known to miss light rainfall events over subtropical and high-latitude oceans (Berg et al., 2010), while simulated precipitation over the Southern Ocean over-estimates drizzle when compared with satellite observations (Franklin et al., 2013; Wang et al., 2015). Some of these differences from TMPA are not mirrored by AWAP over Australia, suggesting possible under-estimation of rain days in TMPA over land (e.g., eastern seaboard, southwest Australia) where the gauge network is relatively dense. Despite these considerations, BARRA-R over-estimates the frequency of light rain days when compared with AWAP, notably in the northern and central regions of Australia, and Tasmania. The UM's parameterized convection scheme assumes that there are many clouds per grid box – which is marginal at the BARRA-R's resolution, and thus produces a bias towards widespread precipitation and provides little indication of the areas which could expect larger rain rates (Clark et al., 2016).

For heavy precipitation days, with amounts [10,50) mm, Figure 9(iii) shows greater similarities between BARRA-R, AWAP and GPCC, over land regions such as the southeast coast of Australia and Tasmania, than for ERA-Interim. BARRA-R shows differences from AWAP and GPCC over Australia north of 30°S where the gauge analyses are poorer. Over the tropical ocean, the two reanalyses show more heavy precipitation days than TMPA.

Lastly for the very heavy precipitation days (≥ 50mm) in Figure 9(iv), it is obvious that ERA-Interim does not fully capture the frequency over land in northern Australia, and southeast Asia, whereas BARRA-R is more comparable with the three reference datasets. This agrees with the findings of Jermey and Renshaw (2016) that higher-resolution regional reanalyses show improvement in representing high-threshold events at these spatial scales. Over the ocean, BARRA-R also shows greater rainfall intensity in the tropics than ERA-Interim, but both reanalyses show lower intensity compared to TMPA. These results reflect the deficiency of the parameterized convection scheme in BARRA-R for estimating convective precipitation amounts in this region.

### 3.3.2. Comparison of monthly totals

Figure 10 and Figure 11 compare differences in domain-averaged monthly totals between the reanalyses (BARRA-R and ERA-Interim) and reference data (TMPA and GPCC) over five separate sub-domains between 80 to 180° E. Precipitation over land and ocean are distinguished. Over the tropical ocean between ±10°N [Figure 10, row (i)], the two reanalyses show different shifts in overall differences from TMPA at around 2010, and these shifts are not apparent in the other sub-domains. Across the sub-domains, the variances of the differences are similar between the two reanalyses.

Over tropical land regions, BARRA-R shows much higher totals than others [Figure 11(i)], due to higher precipitation occurring in mountainous terrains in Papua New Guinea (PNG), Indonesia and Sumatra, and relatively small Indonesian islands (see Figure S3 of the Supplementary Material). Other reanalyses and other gridded precipitation products disagree greatly at these locations with few observations and mountainous terrains (e.g., over PNG in Smith et al., (2013)). BARRA-R (and GPCC) also shows markedly higher monthly totals below 39.2° S [Figure 11(v)], than TMPA and ERA-Interim. This is due to higher BARRA-R precipitation estimates on the west coast and Southern Alps of New Zealand, where precipitation is likely underestimated in TMPA.

The UM can produce grid localized high precipitation in BARRA-R, especially in unstable atmospheric conditions over steep orographic slopes. This issue is not unique to the UM but for instance also occurs in the Weather Research and Forecasting model (Gustafson et al., 2014). When the convective parameterization in non-convective resolving models does not stabilize the air column, meteorological events can develop at the smallest resolvable scales in the model, producing unrealistically strong vertical velocities and precipitation; this is known as "grid-point storms" (Scinocca and McFarlane, 2004; Williamson, 2013; Chan et al., 2014). In our cases, the model only produces isolated excessively intense rainfall over the steep topography.

Such storms occur more readily in models with higher horizontal resolutions (Williamson, 2013). As the resolution increases, resolved motions can produce moisture convergence and increase CAPE very rapidly, and the rate at which column instability is produced depends on the scale of moisture and heat convergence. This also tends to occur over tropical land areas, over steep topography, and during the warm seasons, when the atmosphere is unstable and there is sufficient warm moisture supply at the surface. These considerations do not lend themselves to completely explain the observed bias in BARRA-R.

By contrast, BARRA-R shows better agreement with GPCC and TMPA in other sub-domains between 39.2° to 10.0° S [Figure 11(ii-iv)]. Over the land between 23 to 10°S, BARRA-R simulates wetter summer events than observed in TMPA and GPCC from 2011, when Australia was recovering from drought conditions with the onset of La Nina. Between 39 to 23°S, BARRA-R also simulated wetter events over Mt Kosciuszko, Tasmania, and North Island of New Zealand than TMPA after 2014. This over-estimation is however less apparent when BARRA-R is compared with GPCC.

## 4 Discussion and outlook

The recent development of global and regional reanalyses addresses the need for high-quality, increasingly higher resolution, and longer-term reanalyses, accompanied by estimates of uncertainty, within the research and broader user communities. BARRA is the first regional reanalysis that focuses on the Australasian section of the Southern Hemisphere. It is developed with significant co-investment from state-level emergency service agencies across Australia, due to the advantages of deeper understanding of past weather, including extreme events, and especially in areas that have been poorly served by observation networks. The 29-year BARRA reanalysis, which is expected to be completed in 2019, will ultimately represent a collection of high-resolution gridded meteorological data sets with 12 km and 1.5 km horizontal resolution and 10 minutes to hourly time resolution.

In this paper, we describe the BARRA 12 km regional reanalysis – BARRA-R, which is closely related to the Bureau's regional NWP system, although with an updated UM, 4DVar, variational bias correction, and automated station blacklisting systems. BARRA-R covers a significant region of the globe including parts of South East Asia and the eastern Indian Ocean, the southwest Pacific, Australia and New Zealand and assimilates a wide range of conventional and satellite observations that have proven to improve the skill of NWP.

BARRA-R produces a credible reproduction of the meteorology at and near the surface over land as diagnosed by the selected variables. BARRA-R improves upon its global driving model, ERA-Interim, showing better agreement with point-scale observations of 2 m temperature, 10 m wind speed and surface pressure. Results are similar when BARRA-R is compared with MERRA-2. Daily maximum and minimum statistics for 2 m temperature at 5 km resolution are captured in BARRA-R with smaller biases than ERA-Interim. There appear to be shifts in biases, relative to land observation analyses, over Australia amongst all the reanalyses, mirroring changes in SST. This behaviour however does not coincide with known changes to the

forcing data (soil moisture and SST) used in BARRA-R and requires further analysis to be better understood. BARRA-R's 10 m wind fields show lower biases than ERA-Interim and MERRA-2, but the negative bias during strong winds, which is common amongst other reanalyses, remains significant. Altogether, BARRA-R provides good representation of near-surface extremes, which has implications for its uses for energy management, fire risk and storm damages. The bias could be addressed

via post-processing using methods such as those of Glahn and Lowry (1972), and Rose and Apt (2016). More generally, a variety of post-processing methods can further improve the accuracy of BARRA-R data (e.g., Berg et al., 2012; Frank et al., 2018). Our study did not discern clear merits in BARRA-R analysis and forecast, relative to ERA-Interim analysis, for the pressure-level temperature and wind. Further, there is no conclusive explanation for the shifts in 500, 700 and 850 hPa air temperature occurring at 2010, as comparisons with ERA-Interim and MERRA-2 yield mixed results. Other evaluations of the

UM GA6 configuration including tropical cyclones, precipitation, clouds and large-scale flow, are reported in Walters et al. (2017a; 2017b), albeit in global models at coarser spatial resolutions.

Precipitation fields from BARRA-R show similarities with AWAP and GPCC rain gauge analyses over Australia, where it reflects more similar frequency statistics for heavy rain events and annual mean than ERA-Interim. While this is expected from comparing grids with different resolutions, BARRA-R contains more information pertaining to rain events at local scales.

The frequency statistics (of both light and heavy rain days) of the two reanalyses are markedly different from TMPA over regions exterior to Australia. BARRA-R is likely to be positively biased over land in the regions north of 10° S and New Zealand due to higher precipitation estimates concentrated in regions with high or steep topography. This is partly due to the presence of grid-point storms that occur in non-convective resolving models. Alas, the likely underestimation in observations associated with the high elevations poses difficulties to quantify the wet bias. The  characteristics of grid-point storms in terms

of superficial spatial localization, precipitation amount and vertical wind speed, could be detected and screened out via post-processing. This is important as this model artefact affects the analyses of rainfall averages and extremes.

The disagreement with TMPA is also apparent over the oceans, but consensus between satellite-based products generally degrades over higher latitudes, especially over the Southern Ocean (Behrangi et al., 2014). Over the 2003-2016 period, the variability of the monthly precipitation totals is similar amongst the reanalyses, TMPA and GPCC across the domain. Notable

exceptions are a dry shift occurring in BARRA-R during 2010 over the tropical ocean, and wetter summer events over land in northern and southeast Australia, and the North Island of New Zealand after 2014. These coincident shifts in daily maximum 2 m temperature (over Australia), upper-air temperature (across the BARRA-R domain), and tropical precipitation in all the reanalyses suggest larger differences in large-scale synoptic patterns between them after 2010. Given all the above considerations, local evaluation of BARRA-R reanalysis before application is recommended.

Higher resolution models used to downscale BARRA-R could alleviate the observed shortcomings by resolving sharp topographical features, resolving sub-grid processes (e.g., convection), and using science configurations more suited for a

given climatic region. Assessment of the UM's first Regional Atmosphere (RA1) science configurations for convective-permitting models, recently concluded in December 2017, distinguishes two different science configurations for mid-latitude and tropical regions (RA1-M and RA1-T respectively). Developments in RA1 have produced improvements to 2 m temperature, 10 m wind speed and precipitation (Bush et al., 2019). Further, it is known that BARRA-R's convection scheme,

involving instantaneous adjustment of cloud fields to changes in forcing (e.g., solar heating, land/sea temperature differences), can lead to unrealistic behaviour at places such as coasts and in time (e.g., incorrect diurnal cycle) (Clark et al., 2016). A companion article will examine the merits from downscaling BARRA-R with convective-scale models.

Finally, BARRA represents an important step in supporting the Bureau's ability to prepare for future reanalysis-related activities such as data rescue and reprocessing of observational data. Future reanalyses could use higher resolution models and

ensemble-based forecast and assimilation systems to quantify uncertainties. They will also benefit from international efforts in reprocessing historical conventional and satellite observations with enhanced quality and/or more accurate uncertainty estimates.

*Code availability.* All code, including the UM (version 10.2), VAR (version 2016.03.0), JULES (version 3.0), OPS (version 2016.03.0), SURF (version 2016.07.0) systems, used to produce BARRA is version-controlled under Met Office Science

Repository Service. Readers are referred to https://code.metoffice.gov.uk/trac/home for access information.

*Data availability.* The first releases of the BARRA-R data set for period 2003-2016 are available for academic use, with subsequent releases planned for mid-2019. Readers are referred to http://www.bom.gov.au/research/projects/reanalysis for information on available parameters and access.

*Competing interests.* The authors declare that they have no conflict of interest.

*Author contribution.* PS, DJ, PFH and CJW conceived and/or designed BARRA. CHS, NE and PS developed the BARRA system with inputs from SR, CF, ID and HZ. CHS and NE performed the production and evaluation. CHS prepared the manuscript with contributions from all co-authors.

**Acknowledgements**

Funding for this work was provided by emergency service agencies (New South Wales Rural Fire Service, Western Australia

Department of Fire and Emergency Services, South Australia Country Fire Service, South Australia Department of Environment, Water and National Resources) and research institutions (Antarctic Climate and Ecosystems Cooperative Research Centre (ACE CRC) and University of Tasmania). Funding from Tasmania is supported by the Tasmanian Government and Australian Government, provided under the Tasmanian Bushfire Mitigation Grants Program.

BARRA-R is set up with assistance from the UKMO reanalysis team (R. Renshaw, P. Jermey, J. Davis) and colleagues (A. Maycock, D. Walters, I. Boutle), and many colleagues at the Bureau of Meteorology (T. Le, I. Bermous, L. Rikus, F. Smith, C. Sanders, J. Lee, G. Dietachmayer, J. Le Marshall, X. Sun, J. Fraser, G. Kociuba, C. Tingwell, H. Zhang), the Commonwealth Scientific and Industrial Research Organisation (CSIRO, M. Dix), National Computational Infrastructure (NCI, D. Roberts).

We thank R. Renshaw for providing the observational data from the UKMO and ECMWF archives; S. Moore and T. Carey-Smith at National Institute of Water and Atmospheric Research (NIWA) for providing additional local observations over New Zealand; R. Smalley and D. Jones for their advice on AWAP; P. May, E. Ebert, A. Dowdy and T. Hirst for their feedback on early drafts of manuscript. BARRA-R uses the ERA-Interim data, provided through ARC Centre of Excellence for Climate System Science (P. Petrelli) at NCI. Many of the observational data sets were provided by ECMWF, UKMO and NIWA. The

radio occultation data (CDR v1.0 beta release) were provided by the Radio Occultation Meteorology Satellite Application Facility (ROM SAF, through K. B. Lauritsen and H. Gleisner, Danish Meteorological Institute) which is a decentralized operational RO processing center under EUMETSAT. RO data are available at http://www.romsaf.org. The BARRA project was undertaken with the assistance of resources and services from NCI, which is supported by the Australian Government.

ERA-Interim can be retrieved from ECMWF, https://www.ecmwf.int/en/forecasts/datasets/archive-datasets/reanalysis-
datasets/era-interim. AWAP data can be requested from, http://www.bom.gov.au/climate, TMPA v7 data is retrieved via NASA Goddard Earth Sciences (GES) Data and Information Services Center (DISC), https://disc.gsfc.nasa.gov/datasets/TRMM_3B42_V7/summary, and GPCC v2018 data is retrieved from Deutscher Wetterdienst.

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

**Figures**

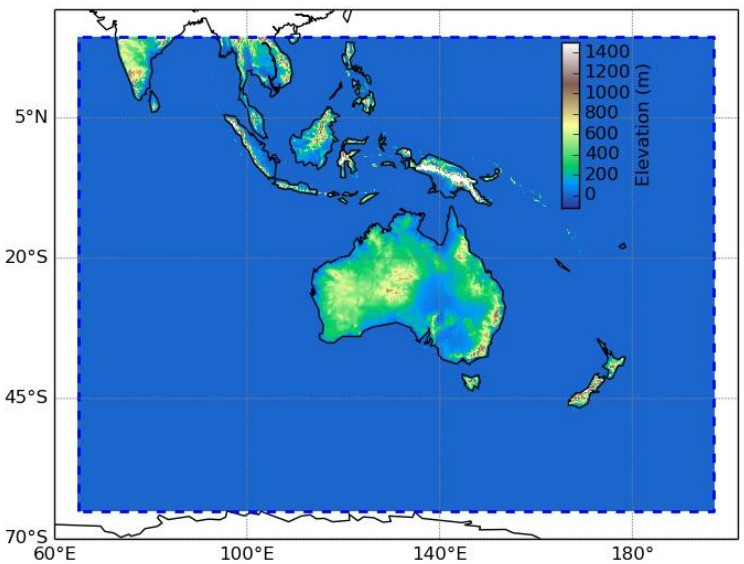

5    **Figure 1  BARRA-R domain enclosed by the dashed box. Blue shading shows the model orography.**

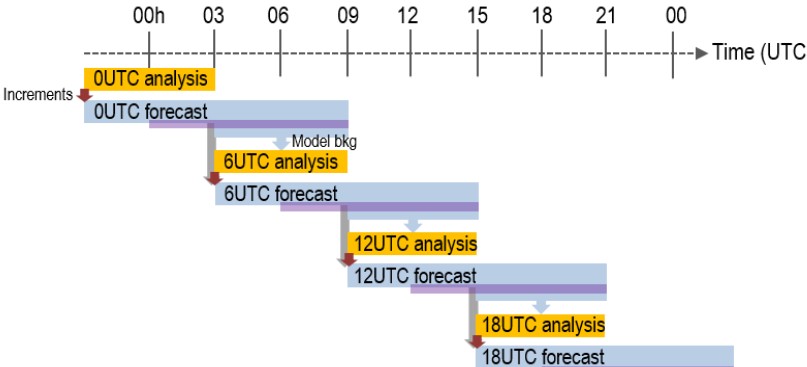

**Figure 2 Cycling setup of BARRA-R at base time t0 = 0, 6, 12, and 18 UTC. Each UM forecast is initialized at t0-3h by the previous forecast (grey arrows) with increments from current analysis (red arrows). The purple bars indicate the time steps of the model states that have been archived.**

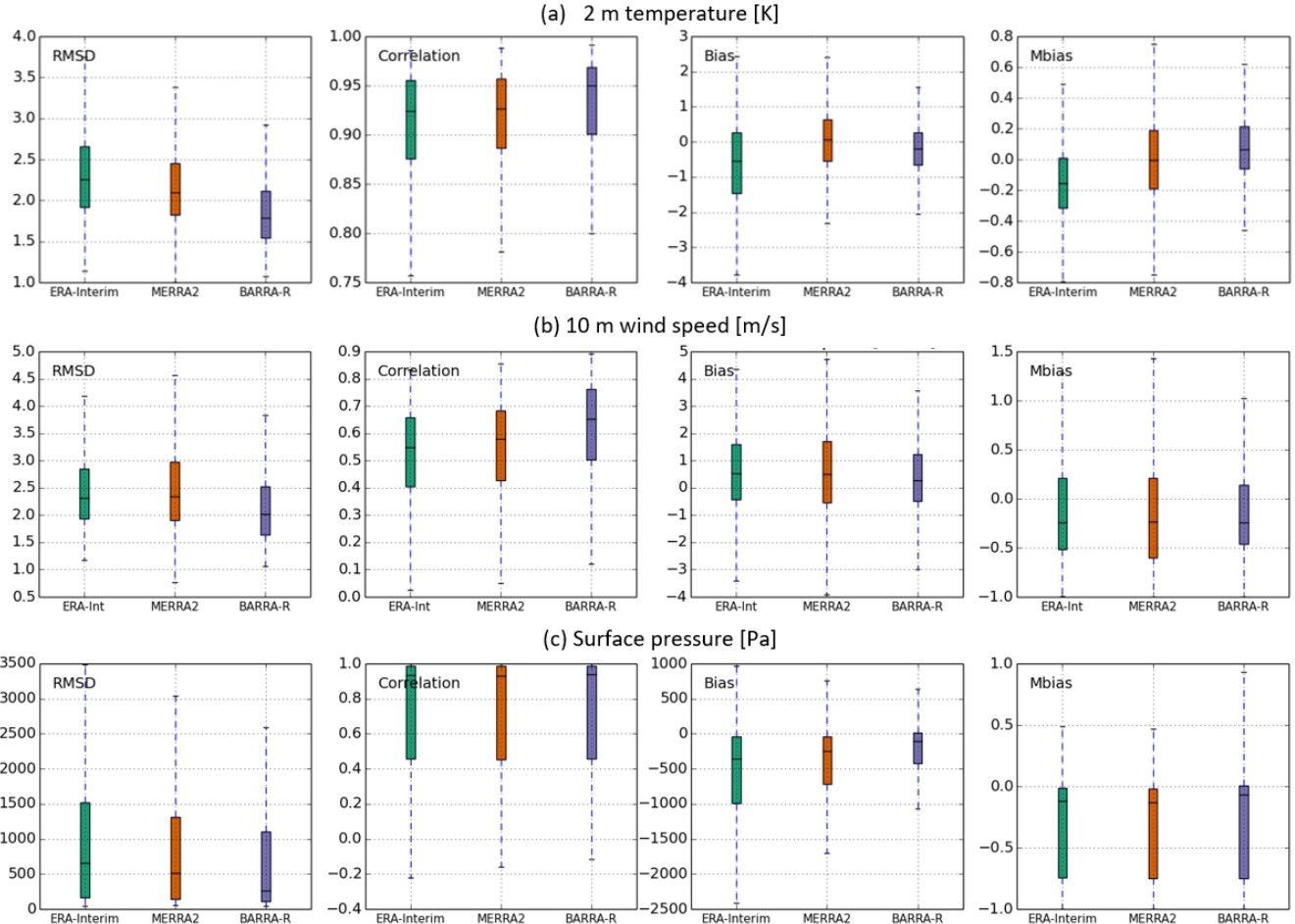

**Figure 3 Boxplots showing the distribution of ERA-Interim, MERRA-2, and BARRA-R evaluation scores for (a) 2 m temperature, (b) 10 m wind speed, and (c) surface pressure over all stations in the BARRA-R domain. The scores are calculated on model forecasts valid between $t_0+5h$ and $t_0+7h$ against observations during 2007-2016. Individual boxes show the interquartile range of the scores, medians are marked in each box and 'whiskers' cover the 5-95% percentile range.**

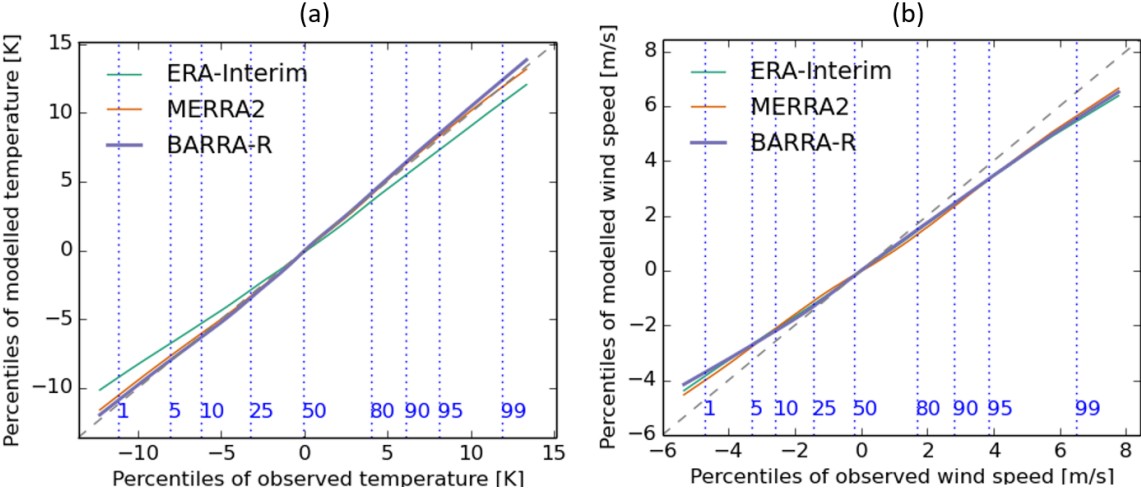

**Figure 4 Comparisons of percentile values between observations and reanalyses for (a) 2 m temperature, and (b) 10 m wind speed during 2010-2013. The values from 0.05% to 99.95% percentiles are calculated using values derived from monthly means. The vertical blue dashed lines indicate the corresponding percentiles of the observations.**

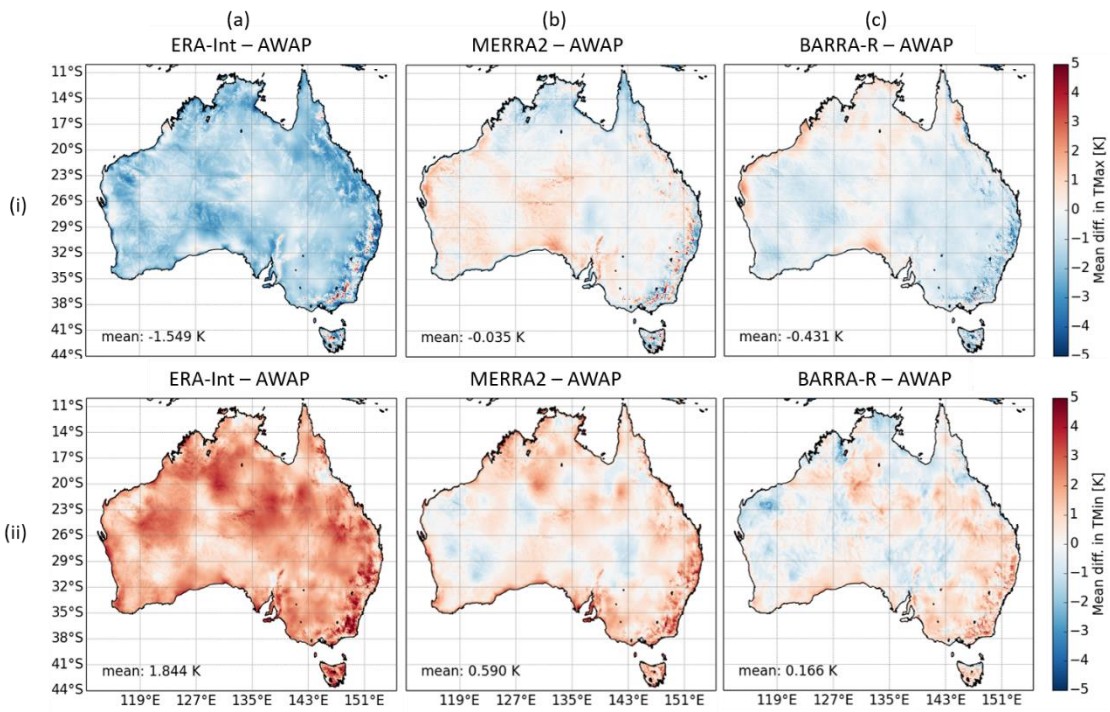

**Figure 5 Mean differences in (row i) daily maximum (TMax) and (ii) minimum (TMin) 2 m temperature [K] for 2007-2016, between (column a) ERA-Interim and AWAP, (b) MERRA-2 and AWAP, and (c) BARRA-R and AWAP. The spatial means of the differences are reported in the text.**

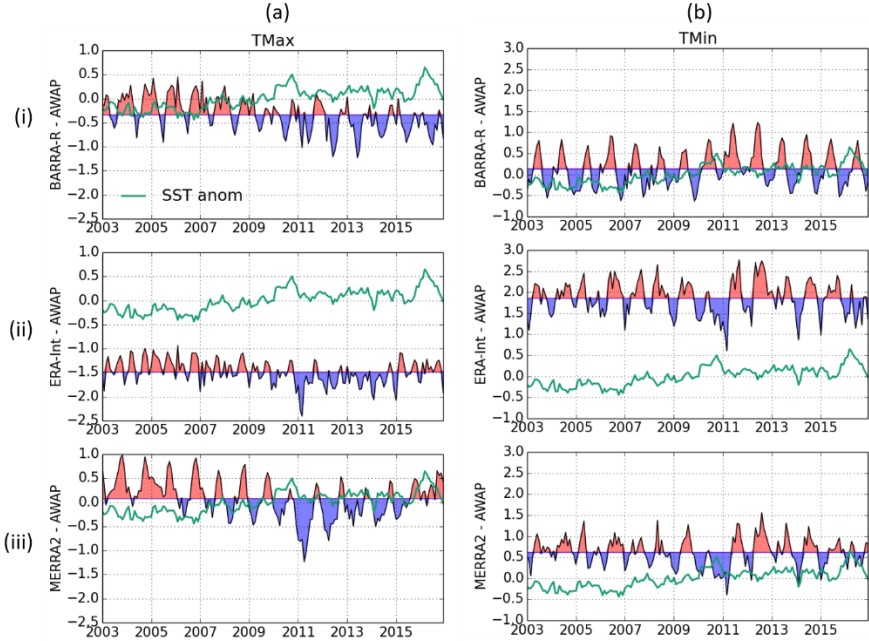

**Figure 6 Monthly mean differences in daily (column a) maximum (TMax) and (b) minimum (TMin) 2 m temperature [K] averaged over Australia, between (row i) BARRA-R and AWAP, (ii) ERA-Interim and AWAP, and (iii) MERRA-2 and AWAP. Black curves are shaded around the 14-year means. Green curves plot the monthly anomalies, from 2003-2016 monthly averages, of the OSTIA sea surface temperature averaged over 46-4° S and 94-174°E.**

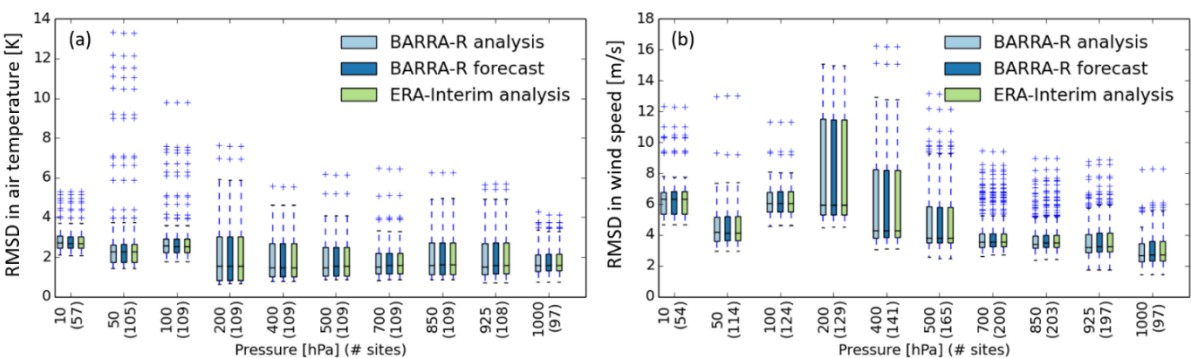

**Figure 7 Boxplots showing the RMSD distribution of BARRA-R $t_0+6$ forecast and $t_0$ analysis, and ERA-Interim analysis for (a) temperature and (b) wind speed at over multiple sites in the BARRA-R domain. RMSD is calculated for temperature and wind speed at pressure levels 10, 50, 100, 200, 400, 500, 700, 850, 925 and 1000 hPa against pilot balloon and radiosonde observations at 0 and 12 UTC. The numbers of sites are indicated in the brackets.**

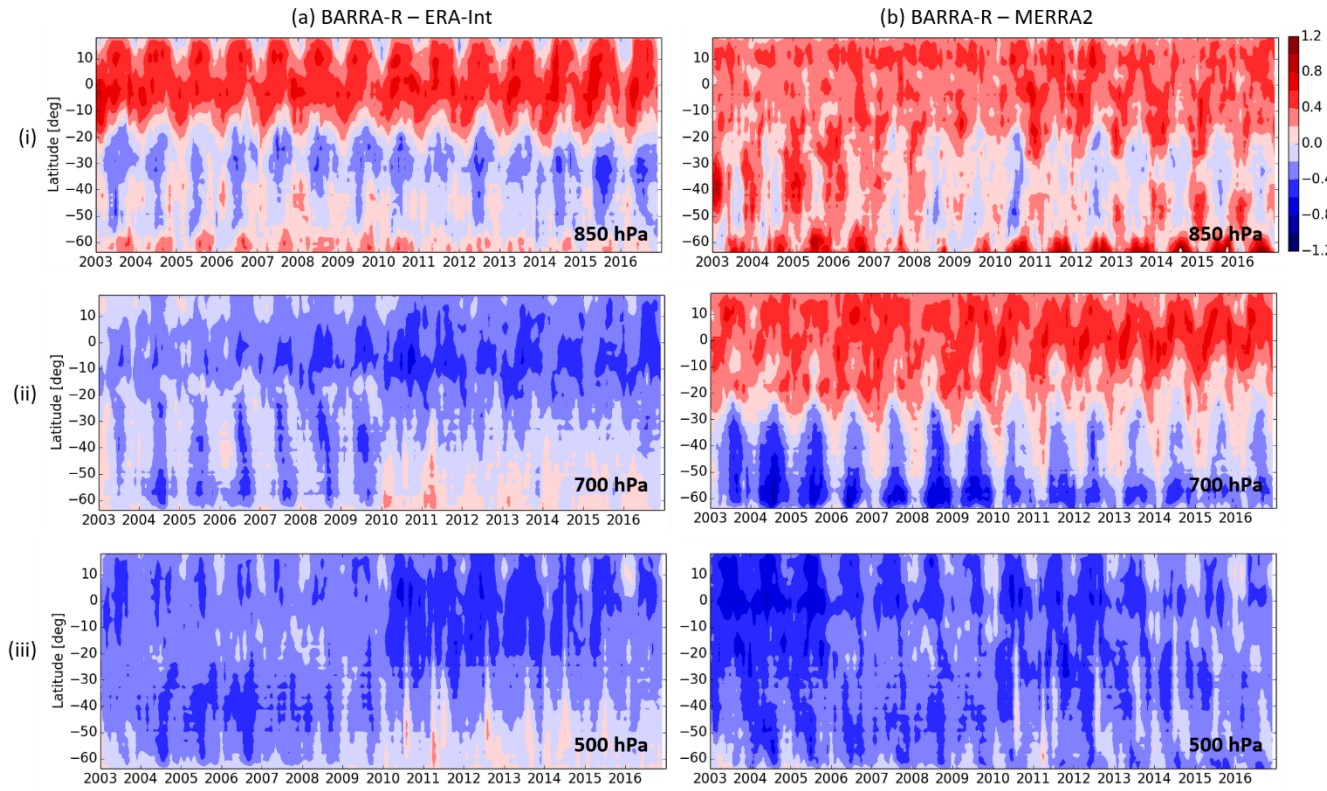

**Figure 8 Hovmöller plots of the monthly difference in zonal mean air temperature [K] at 0 UTC and three pressure levels (row i) 850, (ii) 700, and (iii) 500 hPa, between (column a) BARRA-R and ERA-Interim, and (b) BARRA-R and MERRA-2.**

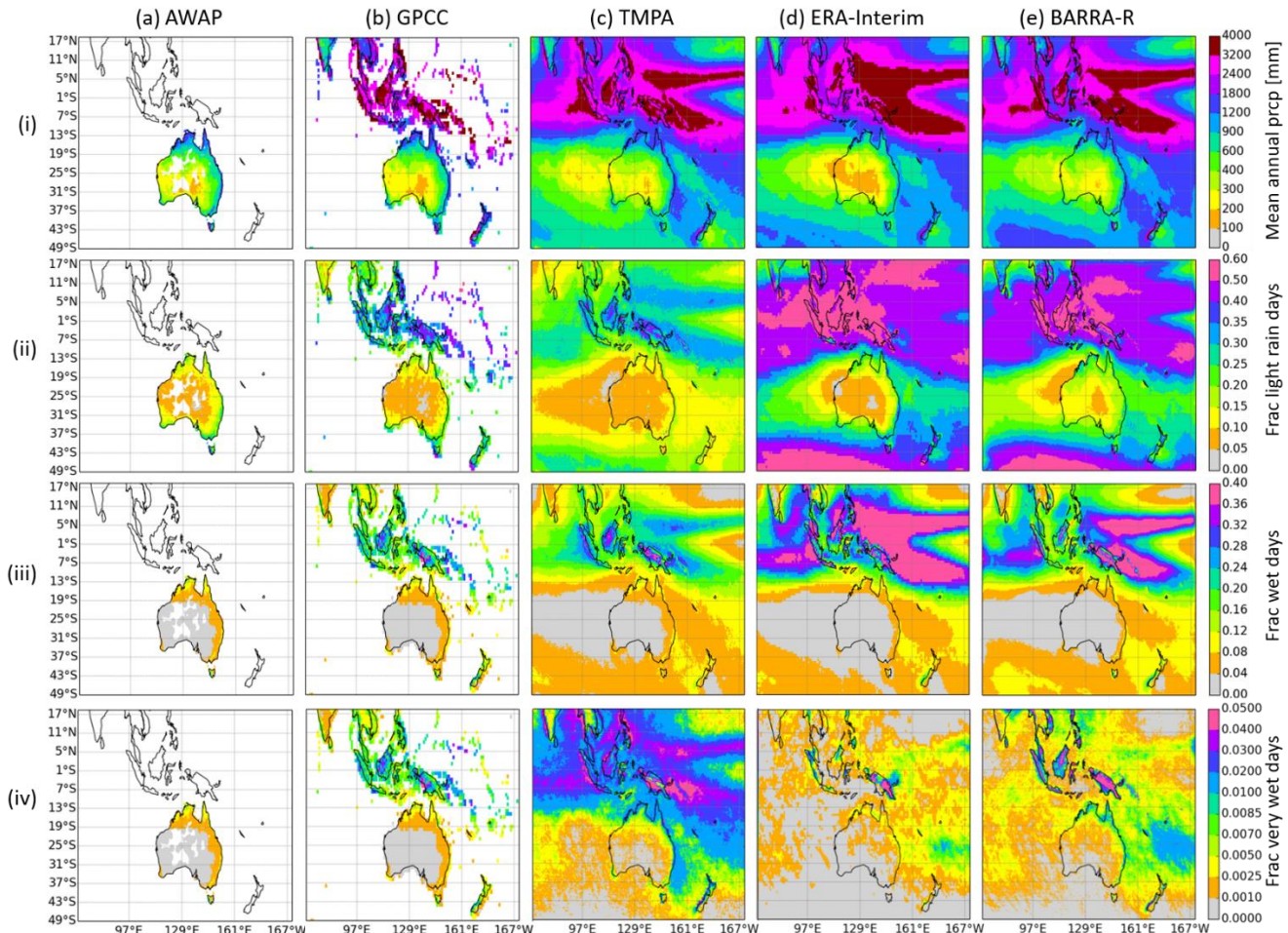

**Figure 9 (row i) Mean annual precipitation [mm], and (ii) fractions of light rain days with 1-10 mm precipitation, (iii) heavy precipitation days with 10-50 mm and (iv) very heavy precipitation days with > 50 mm, over 2007-2016 from (column a) AWAP, (b) GPCC, (c) TMPA, (d) ERA-Interim, and (e) BARRA-R. Regions with more than 10% missing values in AWAP are masked. Close ups of the plots over Australia are provided in the Supplementary Material (Figure S2).**

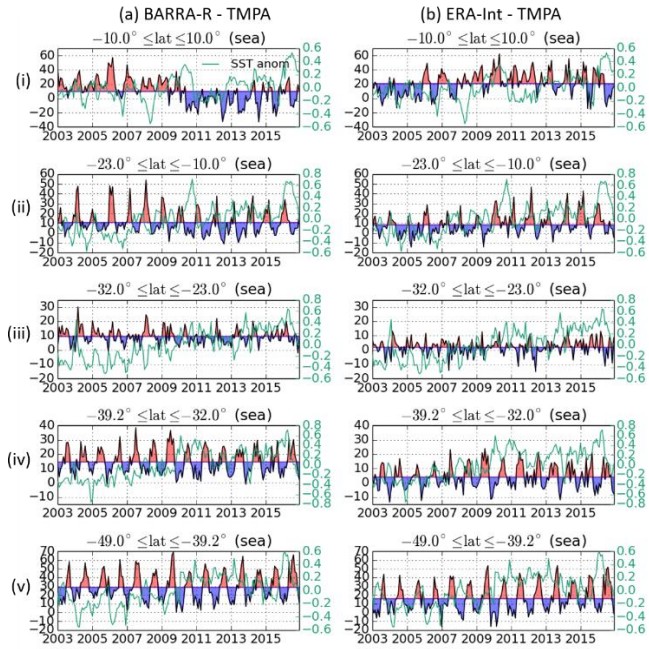

**Figure 10 Differences in monthly precipitation total [mm] averaged over the ocean in five sub-domains (row i-v), between (column a) BARRA-R and TMPA, and (b) ERA-Interim and TMPA. Black curves are shaded around the 14-year means. Green curves plot the monthly anomalies, from 2003-2016 monthly averages, of the OSTIA sea surface temperature averaged over respective sub-domains.**

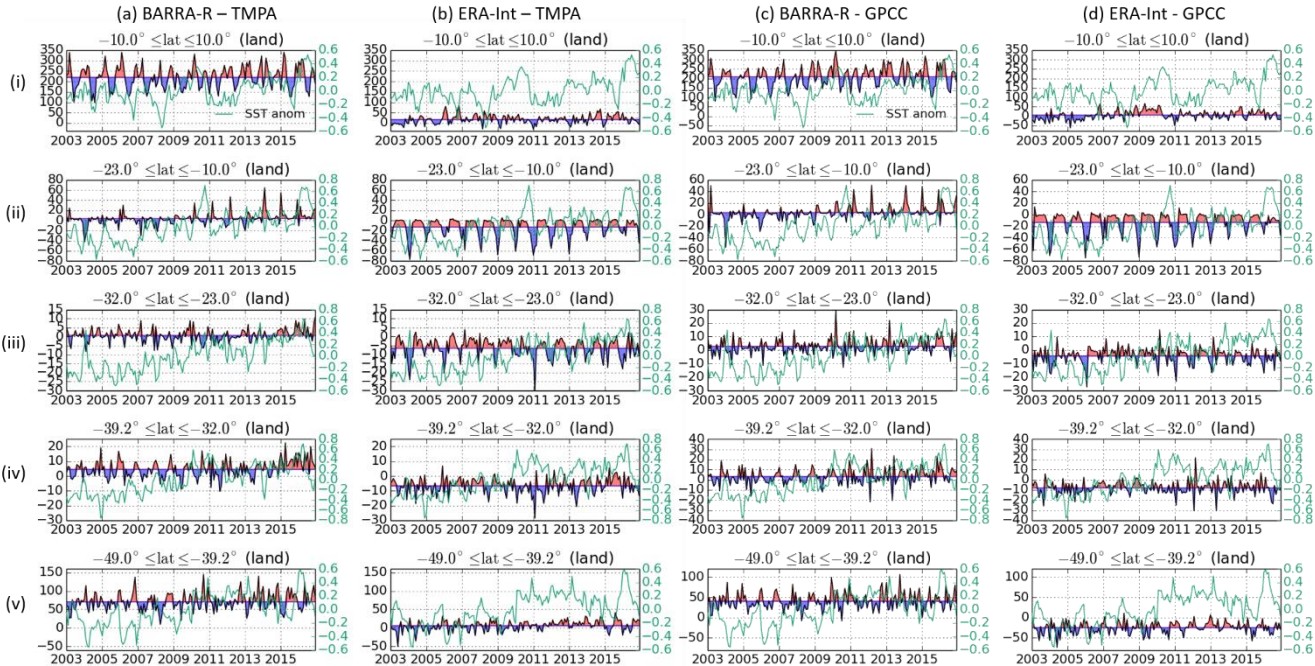

**Figure 11 As with Figure 10(column a) and (b), but over land. Additional comparisons are made between (c) BARRA-R and GPCC, and (d) ERA-Interim and GPCC.**

**Tables**

| Observations | Variables | Time periods | Sources |
|---|---|---|---|
| Land synoptic observations (LNDSYN) | Surface pressure, temperature, humidity, wind | 1978-2018 | Reanalysis prior to 2003 uses the data from ECMWF archive collected for ERA-Interim and ERA-40. Reanalysis between 2003 and 2009 uses the data from ECMWF operational archive. Reanalysis from 2017 uses satellite radiance data from the UKMO operational archive. Reanalysis from 2010 also uses satellite data from the Bureau's operational archive. Bureau's archive also provides 10 minute land synoptic data from 2001, METARS between 2000 to 2009, TEMP from 2002 and WINPRO from 2010. New Zealand National Climate Database (CliDB) provides additional LNDSYN data over New Zealand. |
| Meteorological airfield reports (METARS) | | | |
| Ship synoptic observations (SHPSYN) | | | |
| Buoy | Surface pressure, temperature, wind | | |
| Radiosondes (TEMP) | Upper-air wind, temperature, humidity | 1978-2009 | |
| Wind profilers (WINPRO) | | | |
| Wind-only sondes (PILOT) | Upper-air wind | 1978-2018 | |
| Aircraft Meteorological Data Relay (AMDAR) | Flight-level temperature, wind | 1978-2018 | |
| Air Report (AIREP) | | | |
| Advanced Infrared Sounder (AIRS) | Infrared radiances | 2003-2018 | |
| Advanced TIROS operational vertical sounder (ATOVS) | HIRS/AMSU radiances | 1998-2018 | |
| TIROS operational vertical sounder (TOVS) | MSU and HIRS radiances | 1979-2002 | |
| Infrared Atmospheric Sounding Interferometer (IASI) | Infrared radiances | 2007-2018 | |
| ESA Cloud motion winds (ESACMW) | Satellite radiometer-based winds (satwinds): cloud motion winds, AMV | 1982-2018 | |
| Geostationary Operational Environmental (GOESBUFR) | | 1995-2018 | |
| Meteosat 2nd Generation satellite winds (MSGWINDS) | | 1982-2018 | |
| Japanese Geostationary satellite winds (JMAWINDS) | | 1987-2018 | |
| MODIS winds (MODIS) | | 2005-2018 | |
| SeaWinds | Scatterometer-based winds (scatwinds) | 1996-2009 | |
| Advanced Scatterometer (ASCAT) | | 2007-2018 | |
| GPS Radio Occultation (GPSRO) | Bending angle | 2001-2018 | Reanalysis prior to 2010 uses data provided by Radio Occultation Meteorology Satellite Application Facility (ROM SAF) archive, under EUMETSAT. Reanalysis from 2010 uses the data from the Bureau's operational archive. |
| Australian locally derived satwinds | AMV | 2002-2018 | Bureau of Meteorology operational archive |
| WindSat | Scatwinds | 2015-2018 | |
| Advanced Technology Microwave Sounder (ATMS) | Microwave radiances | 2014-2018 | |
| Cross-track Infrared Sounder (CrIS) | Infrared radiances | 2014-2018 | |
| Tropical Cyclone track (TCBOGUS) | Central pressure and position | 1848-2018 | The International Best Track Archive for Climate Stewardship (IBTrACS) provides the track data up to 2017. The Australian Tropical Cyclone Database is used for 2018. |

**Table** 1   **Observations assimilated in BARRA. Only the period concurrent with the reanalysis period is used. The various data sets were retrieved during the production, and thus the exact periods of each set used may differ.**