# Peer review of "BARRA v1.0: The Bureau of Meteorology Atmospheric high-resolution Regional Reanalysis for Australia"

_Geoscientific Model Development, 2018_

## Referee Comment (RC1) · Anonymous Referee #1 · 9 Jan 2019

The manuscript "BARRA v1.0: The Bureau of Meteorology Atmospheric high-resolution Regional Reanalysis for Australia" provides a description of a novel regional reanalysis system covering Australia, New Zealand and Southeast Asia. Comparison of meteorological parameters from BARRA-R with observations and the forcing global reanalysis ERA-Interim shows that the regional reanalysis provides sound estimates of the atmospheric state and exhibits clear advantages over ERA-Interim, especially in the case of near-surface parameters.

The manuscript is well-written and clearly structured. It provides an introduction on regional reanalysis, a sound description of the model and data assimilation systems

with references to more detailed descriptions with respect to single components of the system and a brief first look into the performance of the system.

My only issue is that the comparison with global reanalyses is only performed against ERA-Interim. I think it would enhance the significance of the results if the authors could include comparisons with the MERRA-2 and JRA-55 reanalyses (see also first minor comment). It would also help to identify biases in BARRA-R originating from the forcing ERA-Interim reanalysis.

Minor comments:

Page 1 Line 19: "[...]Âăthan leading global reanalyses." This is kind of a stretch, as the manuscript only contains comparisons against the global reanalysis ERA-Interim.

Page 2 Line 6: "The latter is currently replaced [...]" instead of "The latter will be replaced [...]"

Page 2 Line 16: "The first [...]" instead of "One of the earliest [...]"

Page 2 Line 19: "[...] as in the global ones [...]" instead of "[...] as the global ones [...]"

Page 3 Line 22: "[...] downscaling [...]" instead of "[...] downscaled reanalysis [...]"

Page 8 Line 12ff: I am a little curious about how well the soil moisture is represented in BARRA-R. In the mid-latitudes, such a system would be started in the late fall or early winter, when the soil is saturated with moisture. However, I can see that for Australia there are a lot of arid or semi-arid regions where such an approach is not feasible. It would be nice if the authors can elaborate on that as soil moisture is not shown in the results but is an important parameter (memory effect) especially in the context of reanalysis compared to NWP.

Page 10 Line 7ff: I understand why it is important to compare the results against the forcing global reanalysis (ERA-Interim). However, as mentioned above, I think it would be good to include comparisons against MERRA-2 and JRA-55 also. On the one hand,
it would help to identify errors or biases in BARRA-R originating from the forcing ERA-Interim, and on the other hand it can be used to look into the confidence of the verifying gridded data sets which might also be error prone.

Page 10 Line 10: "[...] the model levels [...]" instead of "[...] the model's model levels [...]"

Page 12 Line 27: "[...] will be more erroneous [...]" instead of "[...] will be erroneous [...]"

Page 12 Line 28f: I suggest the sentence to be "The accuracy of BARRA-R is expected to worsen during the warm season and at low latitudes, and to improve during the cooler season and at high latitudes where non-convective precipitation is dominant." instead of "The accuracy of BARRA-R is expected to poorer during the warm season and at low latitudes, while better during cooler season and at high latitudes where non-convective precipitation is dominant."

Page 13 Line 6: A comparison over land could also be made with the GPCC data set which is quality controlled and also contains a lot of additional non-public data.

Page 13 Line 11: "row" instead of "column"

Page 13 Line 23: "amounts" instead of "amount"

Page 14 Line 3: "amounts between" instead of "amount"

Page 14 Line 15: Did the authors really see so many "grid-point storms" in the data? Could it also be a misrepresentation / unresolved scale in TMPA?

Page 16 Line 2f: An example of successful statistical post-processing for error correction of regional reanalysis (albeit for radiation) is provided in Frank et al. 2018 (in Solar Energy).

Page 16 Line 4f: The reference should probably (also) be provided in section 3.

Figure 5: Perhaps it would be good to only show the difference plots for ERA Interim and BARRA-R. The color table for the absolute values has too many levels to be useful.

Figure 7: The comparison of AWAP with the other data sets would be easier if all data are plotted with the same domain.

---

## Referee Comment (RC2) · Anonymous Referee #2 · 9 Jan 2019

GENERAL COMMENTS

The topic of the paper is very interesting with the description of the Australian regional reanalysis system BARRA. It gives a good general overview of the system and provides a snapshot of the first results. Therefore, it is a valuable contribution, though there are some shortcomings, which should be addressed in its final version. Particularly, I see as a major problem that there is a major numerical problem in the dataset, which is manifested in "grid-point storms". This should not happen, since reanalysis should be based on a robust and mature numerical weather prediction system, where such problems are already corrected. Overall I propose to accept the paper for publication

provided the comments below are properly addressed in the updated version.

SOME SPECIFIC COMMENTS

The main question for a regional reanalysis is to clearly demonstrate whether the use of such system is justified, which means that more value can be added to the global reanalysis then it would be the case with a pure dynamical downscaling. For this question one has to understand the additional information brought into the reginal system in terms of more precise dynamical and physical description of the atmosphere, but also in terms of additional and advanced use of observations. I miss a summary of this kind from the manuscript though some of these aspects are highlighted here and there in the paper.

Special attention is needed, when different types of driving datasets are used for the reanalysis during the reanalysis period since the continuity should be ensured. Otherwise, there might be some spurious climate signal which is coming from the change of the dataset and not from the climate itself. It should be assessed carefully. In the manuscript we got some examples, as the applied soil moisture fields, which are different from 2015 onwards or the SST and SIC, which is changing in 2007. Some text is needed to emphasise and discuss this aspect.

When the reanalysis is validated sometimes it is not clear enough what are the shortcomings of the datasets used for validation. In some cases it is not clear if the deficiencies identified are coming from the weaknesses of the applied observational datasets or the reanalysis itself. Therefore, at the validation part, it should be also mentioned what are the limitations, which are coming from some of the shortcomings of the validating data itself.

The existence of the "grid-point storms" is embarrassing since such numerical problems should not happen in a reanalysis, where a robust and properly (thoroughly) tested NWP system should be used. Normally, the reanalysis should not be run if such problems are not yet solved. There is a need for a thorough explanation how
this could happen and how this deficiency compromises the validity of the reanalysis results.

The figure captions should provide enough details that the figure can be read without consulting with the main text. In some figures (particularly figure 5 and 7) some information is missing in the caption.

SOME DETAILED PRACTICAL CORRECTIONS

Abstract:

"BARRA-R improves upon ERA-Interim global reanalysis in several areas at point scale to 25km resolution"; would you explain what do you exactly mean and make the sentence clearer, please? (I guess this sentence means that BARRA gives extra value to ERA-Interim until 25km resolution scale.)

Introduction:

page 2, line 6: ERA5 (no hyphen)

page 2, line: 18: please provide reference for Europe reanalysis

page 2, line 25: please give reference for the Copernicus reanalysis

page 3, line 7: "intermittency and covariability"; what do you mean exactly?

The BARRA-R reanalysis:

page 4, line 5: I think, there is a bit of confusion with reference to the UERRA (Uncertainties in Ensembles of Regional Reanalyses) system. UERRA is a system of ensemble of reanalyses, which includes several regional reanalyses. For instance, the UKMO system is one member of the UERRA system, but that is not the "UKMO's UERRA system", since UERRA is the ensemble system itself. I think, the use of UERRA in that regard is not sufficiently precise here and also other parts of the manuscript.

page 6, lines 7-14: the soil moisture treated differently before and after 2015, which

might cause a discontinuity. This should be mentioned and discussed in the text.

page 6, lines 20-30: here inhomogeneity of the SST and SIC fields are revealed and mentioned, but it is not clear how the user should interpret this inhomogeneity while using the BARRA reanalysis dataset.

page 7, line 14: "linearised about a guess", not "linearised around a guess"?

page 8, line 13-15: the one-month spin-up period seems quite short for me particularly for the surface processes.

page 8, lines 22-24: "Most of the observations prior to 2003 are supplied by ECMWF and those between 2003 and 2009 and conventional data from 2003 are extracted from the UKMO operational archives" Does this choice causes any discontinuity? (Though changes in the observing system are usual and the data assimilation system should be able to handle to minimise such discontinuities.)

page 9, line 4: "UKMO's OPS system", what is that? Please have a reference there.

page 9, lines 15-19: changes in observation quantities and screening (thinning) are mentioned here, which might cause discontinuities. It should be explained how this problem can be alleviated.

Preliminary evaluation of ten-year regional reanalysis:

page 9, lines 26-28: it is a basic characteristic of the data assimilation system that the analysis draws nearer to the observations than the guess (RMSE and biases are smaller). This is more a basic sanity check than a real evaluation. On top of that it does not ensure that there are no spurious jumps in standard deviations and biases in the timeseries.

page 10, pages 3-5: this kind of information, what needs to be consolidated and presented altogether in the overview part of the document in order to get an impression what additional information is provided from the regional reanalysis (see also at specific

comments)

page 10, lines 8-9: what observations do you mean, which are not used in the analysis? Or do you mean that since it is a forecast it is an indirect relation, only?

page 11, lines 11-14: please mention "figure not shown" when you discuss a particular aspect, which is not supported by any presented figures.

page 11, lines 16-19: it would be good to include some information about the strengths and weaknesses of the AWAP dataset, which is used for validation since it influences the interpretation of the results.

page 11, lines 24-26, Figure 5: ". . .. BARRA-R shows cold and warm biases around 1K in daily maximum and minimum temperature, respectively. . ." It is quite confusing since the order of the figures is the opposite than in the text (top: minimum, bottom: maximum temperature) and the colour code in the inset is counter-intuitive (red: cold bias, blue: warm bias). I think, that the text states just the opposite than what is seen in the figure: in the minimum temperature there is a cold bias (the reanalysis is colder than the AWAP dataset) and a warm bias (the reanalysis is warmer than the AWAP dataset) in the maximum temperature. Please, clarify this confusion and improve the colour scale of the Figure.

page 12, lines 14-19: one would expect that at least at lower levels some improvements can be detected. Maybe some discussion about this can be put here.

page 13, line 2: ". . . to adjust to remove the excess", is that correct English-wise? Please check.

page 13, line 11: I think, there is a typo there, since it should read as "The first row. . .." and not the "first column".

Figure 7: it is difficult to read these figures, maybe figures only for Australia (as in the supplementary material) would be sufficient.

page 13, line 25: it is very difficult to see the details in Tasmania, I am not sure if such details should be discussed here.

page 14, line 17: what do you mean "temperate region", please? How the latitude bands were selected (for instance 39.2 looks a bit strange choice)?

page 14, lines 18-24: "grid-point storms"; as I mentioned above, I see this numerical issue as a major problem and the reanalysis should not have such numerical instability (and this should have been corrected prior to production).

Summary and outlook:

page 15, line 2: what do you mean on "global effort"?

page 15, line 18: I think the wind bias should be improved with model development efforts and not via post-processing. It is true that some specific users might use post-processing, but I think the modelling community should aim to improve the reanalysis with the improvements of the NWP methodology.

page 15, line 22: what is "GA6 configuration"?

page 16, line 3: I disagree that the grid-point storms should be screened out via post-processing. This is a major numerical problem, which has to be solved before producing a reanalysis.

page 16, line13-14: it is important to get an overview in this paper about the relative merits between reanalysis and downscaling, since this gives justification for having reanalysis instead of simple downscaling. Therefore, some information about this issue should be provided at an early part of this paper.

page 16: lines 18-22: this part of the text is mostly repetition (see page 15, lines 3-7), which should be avoided.

---

## Referee Comment (RC3) · Anonymous Referee #3 · 9 Jan 2019

The paper is well written and seems to contain almost no typos or errors in its discussion and references. It describes a new extensive data set which is very valuable for the region in question and is novel in this way. The results are quite well described and documented. The evaluation methodologies are well thought through and made in a careful way.

I have only some comments and questions as below:

Spin-up period of one month seems short comparing other reanalyses and the reference to Renshaw 2013 may not be so valid today? It would be worthwhile to revisit this point if at all possible, by allowing longer spinup or comparing the first year/season with

a run that has had several years. The question is mainly about soil properties which at least at higher latitudes have a long time constant.

Page 12, line 27: will be erroneous → will be more erroneous

Page 13, line 29: TMPA over land where the rain gauge data . . .. : does TMPA use rain gauge data ? Probably but you describe as multi-satellite analysis; please clarify

---

## Referee Comment (RC4) · Anonymous Referee #4 · 14 Jan 2019

This paper describes the development of the BARRA system for producing atmospheric reanalyses over the Australasian region. The manuscript is well written and the topic is of interest to the atmospheric modelling community. However, I believe the manuscript could be slightly improved with some minor changes.

1) (p 3) The authors refer to a seamless model system in the introduction. Given that there are obvious differences between how the reanalysis, NWP and (presumably) regional climate modelling systems are configured, could the authors explain more clearly the advantages of a seamless modelling system?

2) (section 2) Could the authors provide a short description on the computational re-

quirements of the BARRA system? For example, could the authors comment on how many cores where used and at what rate can reanalysis data be generated?

Interactive
comment

---

## Author Comment (AC1) · 18 Mar 2019

**Response to Reviewer #1**

**1. Response to general comment**

*The manuscript "BARRA v1.0: The Bureau of Meteorology Atmospheric high resolution Regional Reanalysis for Australia" provides a description of a novel regional reanalysis system covering Australia, New Zealand and Southeast Asia. Comparison of meteorological parameters from BARRA-R with observations and the forcing global reanalysis ERA-Interim shows that the regional reanalysis provides sound estimates of the atmospheric state and exhibits clear advantages over ERA-Interim, especially in the case of near-surface parameters. The manuscript is well-written and clearly structured. It provides an introduction on regional reanalysis, a sound description of the model and data assimilation systems with references to more detailed descriptions with respect to single components of the system and a brief first look into the performance of the system.*

*My only issue is that the comparison with global reanalyses is only performed against ERA-Interim. I think it would enhance the significance of the results if the authors could include comparisons with the MERRA-2 and JRA-55 reanalyses (see also first minor comment). It would also help to identify biases in BARRA-R originating from the forcing ERA-Interim reanalysis.*

[AR] We thank the reviewer for providing this useful and thorough review. To further demonstrate the significance of the results, the revised manuscript now includes MERRA-2 in near surface evaluation (revised Figure 3-6), and in comparing the temperature at pressure levels (Figure 8). In the former, we find that BARRA-R shows better agreement with the various reference data than MERRA-2, albeit to lesser degree than ERA-Interim. Thus, this additional comparison does not change the conclusions drawn from this analysis. In the latter, comparisons with ERA-Interim and MERRA-2 show mixed results. We have also included GPCC when evaluating precipitation fields from BARRA-R, particularly exterior to Australia that is not served by AWAP data.

**2. Responses to minor comments**

*2.1. Page 1 Line 19: "[...] than leading global reanalyses." This is kind of a stretch, as the manuscript only contains comparisons against the global reanalysis ERA-Interim.*

[AR] Agreed, and thus MERRA-2 is now included.

*2.2. Page 2 Line 6: "The latter is currently replaced [...]" instead of "The latter will be replaced [...]"*

[AR] Agreed.

*2.3. Page 2 Line 16: "The first [...]" instead of "One of the earliest [...]"*

[AR] Agreed.

*2.4. Page 2 Line 19: "[...] as in the global ones [...]" instead of "[...] as the global ones [...]"*

[AR] Agreed.

*2.5. Page 3 Line 22: "[...] downscaling [...]" instead of "[...] downscaled reanalysis [...]"*

[AR] Agreed.

*2.6. Page 8 Line 12ff: I am a little curious about how well the soil moisture is represented in BARRA-R. In the mid-latitudes, such a system would be started in the late fall or early winter, when the soil is saturated with moisture. However, I can see that for Australia there are a lot of arid or semi-arid regions where such an approach is not feasible. It would be nice if the authors can elaborate on that as soil moisture is not shown in the results but is an important parameter (memory effect) especially in the context of reanalysis compared to NWP.*

[AR] Dharssi and Vinodkumar (2017) found the offline JULES simulations have good skill for the Australian region when validated against ground-based soil moisture observations. BARRA-R also benefitted from the fact that the offline simulation used a 10-year long spin-up period from 1980 so that BARRA-R does not need to start from a particular season. To clarify this in Section 2.1.2, we added the following text: "The simulation used a 10-year long spin-up period and then was run continuously for the 1990 to 2014 period. The soil moisture analyses are found to have good skill for the Australian region when validated against ground-based soil moisture observations (Dharssi and Vinodkumar, 2017)."

*2.7. Page 10 Line 7ff: I understand why it is important to compare the results against the forcing global reanalysis (ERA-Interim). However, as mentioned above, I think it would be good to include comparisons against MERRA-2 and JRA-55 also. On the one hand, it would help to identify errors or biases in BARRA-R originating from the forcing ERA Interim, and on the other hand it can be used to look into the confidence of the verifying gridded data sets which might also be error prone.*

[AR] Agreed. We have now included MERRA-2 in our assessment. JRA-55 is not included for logistical difficulties. Station-based evaluation of screen temperature, 10 m wind speed and surface pressure from MERRA-2 is added to Section 3.1.1 and Figure 3 and 4. Comparisons of gridded daily max and min screen temperature analyses are expanded in Section 3.1.2 and Figure 5 and 6. Comparisons of 500, 700 and 850 hPa air temperature are made in Section 3.2 and Figure 8.

*2.8. Page 10 Line 10: "[...] the model levels [...]" instead of "[...] the model's model levels [...]"*

[AR] Agreed.

*2.9. Page 12 Line 27: "[...] will be more erroneous [...]" instead of "[...] will be erroneous [...]"*

[AR] Agreed.

*2.10.    Page12 Line28f: I suggest the sentence to be "The accuracy of BARRA-R is expected to worsen during the warm season and at low latitudes, and to improve during the cooler season and at high latitudes where non-convective precipitation is dominant." instead of "The accuracy of BARRA-R is expected to poorer during the warm season and at low latitudes, while better during cooler season and at high latitudes where non-convective precipitation is dominant."*

[AR] Agreed.

*2.11.    Page 13 Line 6: A comparison over land could also be made with the GPCC data set which is quality controlled and also contains a lot of additional non-public data.*

[AR] Agree. Our choice for using AWAP is based on its spatial resolution (5 km) is more comparable with BARRA-R. However, GPCC daily data (version 2018) is of much lower resolution of 1 degree, but

is beneficial for comparison over land outside Australia. Section 3.3, with expanded Figure 9-11, has been expanded to include GPCC comparison.

*2.12.    Page 13 Line 11: "row" instead of "column"*

[AR] Agreed.

*2.13.    Page 13 Line 23: "amounts" instead of "amount"*

[AR] Agreed.

*2.14.    Page 14 Line 3: "amounts between" instead of "amount"*

[AR] Agreed.

*2.15.    Page 14 Line 15: Did the authors really see so many "grid-point storms" in the data? Could it also be a misrepresentation / unresolved scale in TMPA?*

[AR] It is difficult to completely explain the relative contributions, to the observed differences, between the model's grid point storms and unresolved precipitation in reference data. This is because the differences are mainly observed over steep orography and isolated islands (see the new Figure S5 in Supplementary Material), where rain gauge observations are scarce and/or non-representative at areal scale. It is well known in the literature that numerical weather models generate "grid point storms" – an issue not unique to Unified Model (UM). For instance, Weather Research and Forecasting (WRF) model can also generate such storms (Gustafson et al., 2014).

We expand the discussion on this in the manuscript and included one additional illustration (Figure S3 in Supplementary Material):

The text reads "Over tropical land regions, BARRA-R shows much higher totals than others [Figure 11(i)], due to more precipitation occurring at high or sharp topographical regions in Papua New Guinea (PNG), Indonesia and Sumatra, and relatively small Indonesian islands (see the Supplementary Material). Other reanalyses and other gridded precipitation products disagree greatly at these locations with few observations and mountainous terrains (e.g., over PNG in Smith et al., (2013)). BARRA-R (and GPCC) also shows markedly higher monthly totals below 39.2° S [Figure(v)], than TMPA and ERA-Interim. This is due to higher BARRA-R precipitation estimates on the west coast and Southern Alps of New Zealand (see the Supplementary Material), where precipitation is likely underestimated in TMPA.

UM can produce grid localized high precipitation in BARRA-R, especially in unstable atmospheric conditions over steep orographic slopes. This issue is not unique to UM, but, for instance, can also occur in the Weather Research and Forecasting model (Gustafson et al., 2014). When the convective parameterization does not stabilize the air column, meteorological events can develop at the smallest resolvable scales in the model, producing unrealistically strong vertical velocities and precipitation; this is known as "grid-point storms" (Scinocca and McFarlane, 2004; Williamson, 2013). Such storms occur more readily in models with higher horizontal resolutions (Williamson, 2013). As the resolution increases, resolved motions can produce moisture convergence and increase CAPE very rapidly, and the rate at which column instability is produced depends on the scale of moisture and heat convergence. This also tends to occur over the land in the tropics and during the warm seasons when the atmosphere is unstable and there is sufficient warm moisture

supply at the surface. These considerations do not lend themselves to explain the observed bias in BARRA-R."

*2.16.      Page 16 Line 2f: An example of successful statistical post-processing for error correction of regional reanalysis (albeit for radiation) is provided in Frank et al. 2018 (in Solar Energy).*

[AR] Agreed. We added this example to the conclusion. The text now reads "More generally, a variety of post-processing methods can further improve the accuracy of BARRA-R data (e.g., Berg et al., 2012; Frank et al., 2018)."

*2.17.      Page 16 Line 4f: The reference should probably (also) be provided in section 3.*

[AR] Agreed. The reference has been moved to the first paragraph of Section 3.3.

*2.18.      Figure 5: Perhaps it would be good to only show the difference plots for ERA Interim and BARRA-R. The color table for the absolute values has too many levels to be useful.*

[AR] Agreed. Figure 5 has been modified to show mean difference in daily maximum and minimum 2 metre temperature of ERA-Interim, MERRA-2 and BARRA-R from AWAP.

*2.19.      Figure 7: The comparison of AWAP with the other data sets would be easier if all data are plotted with the same domain.*

[AR] Agreed. AWAP plots in Figure 7 have plotted over the same domain. The closeup of the maps over Australia is provided in Figure S1 in the Supplementary Material.

**Other significant changes not described above**
- **Figure 4**, which shows comparisons between models and observations in terms of percentiles, has been revised. An error in the codes to interpolate model grids to observing station locations has been fixed.
- **Table 1**, which lists the various observational data sources, has been revised to reflect the extension of the reanalysis period to include 2018. Data sources has also been corrected.

[revised manuscript text omitted]

**Supplementary Material**

[Figure]

Figure S1 Spatial maps of the difference in RMSD between (left column) BARRA-R and ERA-Interim, and (right) BARRA-R and MERRA-2. RMSD is calculated on model forecasts against observed (a) 2 m temperature, (b) 10 m wind speed and (c) surface pressure. These figures supplementing Figure 3.

[Figure]

**Figure S2 As Figure 8 but over Australia only. (i) Mean annual precipitation [mm], and (ii) fractions of rain days, (iii) heavy precipitation days and (iv) very heavy precipitation days, from 2007 to 2016 from (a) AWAP, (b) GPCC, (c) TMPA, (d) ERA-Interim, and (e) BARRA-R. Regions with more than 10% missing values in AWAP are masked.**

[Figure]

**Figure S3 Mean difference in daily precipitation between BARRA-R and TMPA from 2007-2016 over two sub-domains in the tropics and New Zealand. The data is regridded to the TMPA 0.25° grid.**

|  | O-B | | O-A | |
| --- | --- | --- | --- | --- |
| **Fields** | **Bias** | **RMSD** | **Bias** | **RMSD** |
| Surface temperature [K] | -0.09 | 1.78 | -0.10 | **1.61** |
| Surface pressure [Pa] | -3.67 | 101.69 | **-2.08** | **68.85** |
| Surface relative humidity [%] | 0.0 | 10.0 | 0.00 | **8.0** |
| Surface zonal wind [m/s] | 0.05 | 1.97 | **-0.01** | **1.74** |
| Surface meridional wind [m/s] | 0.04 | 1.94 | **0.01** | **1.72** |
| Aircraft potential temperature [K] | -0.24 | 1.34 | **-0.17** | **1.10** |
| Aircraft zonal wind [m/s] | -0.04 | 3.05 | **-0.03** | **2.09** |
| Aircraft meridional wind [m/s] | -0.18 | 3.06 | **-0.07** | **2.07** |
| Sonde temperature at 980 hPa [K] | -0.15 | 1.11 | **-0.08** | **0.81** |
| Sonde temperature at 500 hPa [K] | -0.33 | 0.92 | **-0.18** | **0.60** |
| Sonde zonal wind at 980 hPa [m/s] | -0.15 | 2.45 | **-0.06** | **1.45** |
| Sonde zonal wind at 500 hPa [m/s] | -0.17 | 2.52 | **-0.07** | **1.41** |
| Sonde meridional wind at 980 hPa [m/s] | 0.23 | 2.34 | **0.09** | **1.38** |
| Sonde meridional wind at 500 hPa [m/s] | 0.11 | 2.44 | **0.03** | **1.39** |
| Satwind zonal wind [m/s] | 0.36 | 3.16 | **0.27** | **2.72** |
| Satwind meridional wind [m/s] | 0.05 | 2.90 | **0.01** | **2.40** |
| Scatwind zonal wind [m/s] | 0.06 | 1.39 | **0.03** | **0.95** |
| Scatwind meridional wind [m/s] | 0.20 | 1.78 | **-0.02** | **1.32** |

**Table S1 Comparisons of the 10-year (2007-2016) mean of the RMSD and bias between the analyses and observations (O-A) and those between the background and observations (O-B), calculated for various observational types across the BARRA-R domain. Bold values show reduction in the RMSD and the magnitude of the bias by the analyses, i.e., the analyses draw the model forecasts closer to these observation types.**

---

## Author Comment (AC2) · 18 Mar 2019

**Response to Reviewer #2**

**1.  Response to general comment**

*The topic of the paper is very interesting with the description of the Australian regional reanalysis system BARRA. It gives a good general overview of the system and provides a snapshot of the first results. Therefore, it is a valuable contribution, though there are some shortcomings, which should be addressed in its final version. Particularly, I see as a major problem that there is a major numerical problem in the dataset, which is manifested in "grid-point storms". This should not happen, since reanalysis should be based on a robust and mature numerical weather prediction system, where such problems are already corrected. Overall I propose to accept the paper for publication provided the comments below are properly addressed in the updated version.*

[AR] We thank the reviewer for their comments. We acknowledge the shortcomings of the work and have used this opportunity to undertake a major revision of the manuscript. In particular, we expanded our discussion around "grid-point storms" to emphasize the difficulties to remove their occurrence completely. We have also extended the analysis period from 2007-2016 to 2003-2016, and conducted additional analyses to ascertain (1) the locations of precipitation excess in the tropics and New Zealand, and (2) whether there are shifts in BARRA-R due to SST and soil moisture changes. See below for more details.

*The main question for a regional reanalysis is to clearly demonstrate whether the use of such system is justified, which means that more value can be added to the global reanalysis then it would be the case with a pure dynamical downscaling. For this question one has to understand the additional information brought into the reginal system in terms of more precise dynamical and physical description of the atmosphere, but also in terms of additional and advanced use of observations. I miss a summary of this kind from the manuscript though some of these aspects are highlighted here and there in the paper.*

[AR] The introduction has reviewed several papers on the usefulness of regional reanalyses over dynamical downscaling, underpinning efforts around the various regional reanalysis projects internationally.
The comparisons of short-ranged O-A (observation – analysis) and O-B (observation – background) statistics in Table 1 (now moved to Table S1 of the Supplementary Material) showed that, with O-A being consistently better O-B for various observational types, an analysis within the BARRA-R system yields a more accurate short-ranged forecast than simply using the background, where the background from the previous analysis is, by extension, better than pure dynamical downscaling from the very first cycle.

**2.  Response to specific comments**
*2.1. Special attention is needed, when different types of driving datasets are used for the reanalysis during the reanalysis period since the continuity should be ensured. Otherwise, there might be some spurious climate signal which is coming from the change of the dataset and not from the climate itself. It should be assessed carefully. In the manuscript we got some examples, as the applied soil moisture fields, which are different from 2015 onwards or the SST and SIC, which is changing in 2007. Some text is needed to emphasise and discuss this aspect.*

[AR] Agreed. We have now extended the study period to include BARRA-R data from 2003-2016 (previously 2007-2016), to consider whether the change of the OSTIA data at 2007 leads to discontinuities. Our evaluation has been extended as follows:

- Section 3.1.2 "Comparisons with gridded analysis of observed 2 m temperature has been extended to look at the 14-year changes to biases in TMax (daily maximum temperature) and TMin (daily minimum temperature) relative to AWAP – in new Figure 6. We found shifts in biases in BARRA-R, but also in the global reanalyses ERA-Interim and MERRA-2. Further, these changes do not coincide with our changes in soil moisture initialization in 2014-2015 or OSTIA data.
- Section 3.2 "Pressure levels" has been extended to look at 14-year changes to air temperature at 500, 700 and 850 hPa pressure levels – new Figure 8. We found shifts in biases between BARRA-R and ERA-Interim from 2010, but these shifts are absent when comparing BARRA-R with MERRA-2. The results are therefore inconclusive.
- Section 3.3 "Precipitation" has been extended to look at 14-year differences in monthly precipitation between BARRA-R and TMPA, ERA-Interim and TMPA, BARRA-R and GPCC, and ERA-Interim and GPCC – revised Figure 10 and 11. The most apparent shifts in precipitation occur in north Australia, where BARRA-R shows wetter summer events after 2010, at the end of the Australian millennium drought and onset of La Nina.

We summarise our observations in the conclusion as follows: "These coincident shifts in daily maximum 2 m temperature (over Australia), upper-air temperature (across the BARRA-R domain), and tropical precipitation in all the reanalyses suggest larger differences in large-scale synoptic patterns after 2010."

*2.2. When the reanalysis is validated sometimes it is not clear enough what are the shortcomings of the datasets used for validation. In some cases, it is not clear if the deficiencies identified are coming from the weaknesses of the applied observational datasets or the reanalysis itself. Therefore, at the validation part, it should be also mentioned what are the limitations, which are coming from some of the shortcomings of the validating data itself.*

[AR] Agree. The first paragraphs of Section 3.1.1 "Point evaluation of 2m temperature, 10 m wind speed, and surface pressure", 3.1.2 "Comparisons with gridded analyses of observed 2 m temperature", and 3.3 "Precipitation" describe the issues of the validating data.

*2.3. The existence of the "grid-point storms" is embarrassing since such numerical problems should not happen in a reanalysis, where a robust and properly (thoroughly) tested NWP system should be used. Normally, the reanalysis should not be run if such problems are not yet solved. There is a need for a thorough explanation how this could happen and how this deficiency compromises the validity of the reanalysis results.*

[AR] The Unified Model is sufficiently robust to be useful for many operational meteorological centres in Australia, UK, India, Singapore, Korea, South Africa and New Zealand. The issue of "grid-point storms" is also not unique to UM but for instance, also occurs in the widely-used Weather Research and Forecasting (WRF) model from NCAR. When the convective (sub-grid) parameterization scheme in non-convective resolving models does not stabilize the air column, meteorological events can develop at the smallest resolvable scales in the model, producing unrealistically strong vertical velocities and precipitation (Scinocca and McFarlane, 2004; Williamson, 2013). The resulting "grid-point storms" occur more readily in models with higher horizontal resolutions (Williamson, 2013). The issue becomes unavoidable for BARRA-R as it aims to be sufficiently higher resolution than global reanalyses but could not be sufficiently high resolution (< 2 km) (and computationally prohibitive) to resolve convection explicitly without the need for a convective parameterization scheme.

Further, we do not think that the wet biases in BARRA-R over the tropics and New Zealand are entirely due to grid point storms. Additional analyses have been made to identify the location of

precipitation excess in the tropics and New Zealand (Figure S3 in Supplementary Material). We found that the higher precipitation in BARRA-R are concentrated at high or sharp topographical regions in PNG, Indonesia, Sumatra and small Indonesian Islands, and west coast and Southern Alps of New Zealand. At these locations, GPCC (gauge analysis) and TMPA would underestimate the precipitation. With these considerations, the actual levels of bias observed in BARRA-R are not entirely clear.

2.4. *The figure captions should provide enough details that the figure can be read without consulting with the main text. In some figures (particularly figure 5 and 7) some information is missing in the caption.*

[AR] Agreed. The captions have been revised.

Figure 5 has been simplified and its caption now reads " Mean differences in (row i) daily maximum (TMax) and (ii) minimum 2 m temperature [K] for 2007-2016, between (column a) ERA-Interim and AWAP, (b) MERRA-2 and AWAP, and (c) BARRA-R and AWAP. The spatial means of the differences are reported in text.".

The caption for Figure 9 (previously Figure 7) has been revised to read "(row i) Mean annual precipitation [mm], and (ii) fractions of light rain days with 1-10 mm precipitation, (iii) heavy precipitation days with 10-50 mm and (iv) very heavy precipitation days with > 50 mm, over 2007-2016 from (column a) AWAP, (b) GPCC, (c) TMPA, (d) ERA-Interim, and (e) BARRA-R. Regions with more than 10% missing values in AWAP are masked. Close ups of the plots over Australia are provided in the Supplementary Material."

2.5. *Abstract: "BARRA-R improves upon ERA-Interim global reanalysis in several areas at point scale to 25km resolution"; would you explain what do you exactly mean and make the sentence clearer, please? (I guess this sentence means that BARRA gives extra value to ERA-Interim until 25km resolution scale.)*

[AR] Agreed. We are careful with specifying the spatial scale of the evaluation as the relative skills between BARRA and ERA-Interim are expected to vary with the scale.

The text has been rewritten as
"BARRA-R provides a realistic depiction of the meteorology at and near the surface over land as diagnosed by temperature, wind speed, surface pressure, and precipitation. Comparing against global reanalyses ERA-Interim and MERRA-2, BARRA-R scores lower root-mean-square errors when evaluated against (point-scale) 2 m temperature, 10 m wind speed and surface pressure observations. It also shows reduced biases in daily 2 m temperature maximum and minimum at 5 km resolution, and a higher frequency of very heavy precipitation days at 5 and 25 km resolution when compared to gridded satellite and gauge analyses."

2.6. *page 2, line 6: ERA5 (no hyphen)*

[AR] Agreed.

2.7. *page 2, line: 18: please provide reference for Europe reanalysis*

[AR] Agree, it now reads "The first regional reanalyses was the North America Regional Reanalysis (NARR, Mesinger et al., 2006), and the more recent examples include the Arctic System Reanalysis (ASR, Bromwich et al., 2018), Indian Monsoon Data Assimilation and Analysis (IMDAA, Mahood et al.,

2018) and Uncertainties in Ensembles of Regional Reanalyses (UERRA) in Europe (Borsche et al. (2015) and therein)."

*2.8. page 2, line 25: please give reference for the Copernicus reanalysis*

[AR] Agreed. I have added a reference to Ridal et al. (2017).

*2.9. page 3, line 7: "intermittency and covariability"; what do you mean exactly?*

[AR] We expand the text, which now reads "For renewable energy production, they can provide valuable information on intermittency (e.g., wind lull) and covariability (e.g., spatial or between variables) of phenomena."

*2.10.     page 4, line 5: I think, there is a bit of confusion with reference to the UERRA (Uncertainties in Ensembles of Regional Reanalyses) system. UERRA is a system of ensemble of reanalyses, which includes several regional reanalyses. For instance, the UKMO system is one member of the UERRA system, but that is not the "UKMO's UERRA system", since UERRA is the ensemble system itself. I think, the use of UERRA in that regard is not sufficiently precise here and also other parts of the manuscript.*

[AR] Agreed. It now reads "…UKMO's system in UERRA". Other appearances have been modified accordingly.

*2.11.     page 6, lines 7-14: the soil moisture treated differently before and after 2015, which might cause a discontinuity. This should be mentioned and discussed in the text.*

[AR] Agree. We have added to Section 2.1.2, which now also reads " The daily initialisation was conducted with the purpose of avoiding spurious drift in the BARRA moisture fields and reducing the time needed to spin up from ERA-Interim initial conditions. However, as multiple parallel production streams are needed to produce the reanalysis (see Section 2.2), a discontinuity in soil moisture in the bottom two layers exists between successive production streams, although soil moisture in the top two layers become stable after one-month of runs. A discontinuity occurring at the 2014-2015 changeover has recently been reported by BARRA data users. These impacts, particularly on forested regions where trees extract water from the deep soil layers, are under investigation."

So far, our investigations described in our reply to comment 2.1 do not show changes to screen temperature, precipitation, and upper-air temperature, due to the changes in soil moisture initialization.

2.12.     page 6, lines 20-30: here inhomogeneity of the SST and SIC fields are revealed and mentioned, but it is not clear how the user should interpret this inhomogeneity while using the BARRA reanalysis dataset.

[AR] Agree. Our investigations of the possible impacts are described in our reply to comment 2.1.

We have included plots of OSTIA SST anomalies in long-term analyses in Figure 6, 10 and 11. They do not show unexpected discontinuities at 2006/2007.

*2.13.     page 7, line 14: "linearised about a guess", not "linearised around a guess"?*

[AR] Agreed.

*2.14.    page 8, line 13-15: the one-month spin-up period seems quite short for me particularly for the surface processes.*

[AR] First, BARRA-R is initialized from ERA-Interim, and not an arbitrary state. Second, the land state and in particular sub surface soil moisture can have a very long memory. One month is indeed likely to be too short to adequately spin-up the BARRA soil moisture. To ameliorate this problem, BARRA soil moisture was initialised using soil moisture analyses from an off-line JULES simulation. This off-line JULES simulation used a 10-year long spin-up period and then was run continuously for the 1990 to 2014 period. Even so, it appears that the soil moisture at bottom two layers did not fully spin up after one-month runs, while the top two layers appear stable after a month. To highlight this shortcoming with the added text in Section 2.1.2 that reads,
"The daily initialisation was conducted with the purpose of avoiding spurious drift in the BARRA moisture fields and reducing the time needed to spin up from ERA-Interim initial conditions. However, as multiple parallel production streams are needed to produce the reanalysis (see Section 2.2), a discontinuity in soil moisture in the bottom two layers exists between successive production streams, although soil moisture in the top two layers become stable after one-month of runs. A discontinuity occurring at the 2014-2015 changeover has recently been reported by BARRA data users. These impacts, particularly on forested regions where trees extract water from the deep soil layers, are under investigation."

*2.15.    page 8, lines 22-24: "Most of the observations prior to 2003 are supplied by ECMWF and those between 2003 and 2009 and conventional data from 2003 are extracted from the UKMO operational archives" Does this choice causes any discontinuity? (Though changes in the observing system are usual and the data assimilation system should be able to handle to minimise such discontinuities.)*

[AR] The observational data are extracted from different archives due to practical reasons (project funding and delivery). Conventional observation data are maintained and transmitted to the international community via GTS (global telecommunication system) by local operators so all meteorological centres share the same data, albeit in different formats in respective data archives. Satellite data is produced using the same algorithms and software packages developed by the satellite operators and their partners.

2.16.    page 9, line 4: "UKMO's OPS system", what is that? Please have a reference there.

[AR] It refers to Observation Processing System. The best reference we can find is Rawlins et al. (2007), with methods described in Lorenc and Hammon (1988). Both are cited.

*2.17.    page 9, lines 15-19: changes in observation quantities and screening (thinning) are mentioned here, which might cause discontinuities. It should be explained how this problem can be alleviated.*

[AR] Changes in observation quantities are unavoidable in any reanalysis, be it from changes made to the screening mid-production or changes in observing systems. While it is unfortunate that we had to make an adjustment to observation screening mid-production, any reanalyses including BARRA need to be used with some caution, due to inherent variability in quality due to observation changes. It is not within the scope of this work to look further. We added a discussion to Section 3.2 that reads, " There are abrupt changes to the amount of satellite data assimilated at the start and end of satellite missions and the various observational data archives; in some cases, changes occur when corrections were made to the observation screening and thinning rules mid-production of the

2010-2015 reanalyses. The impacts of such changes, known to cause artificial shifts and spurious trends in a reanalysis (e.g., Thorne and Vose, 2010; Dee et al., 2011) are still to be investigated for BARRA-R."

*2.18.     page 9, lines 26-28: it is a basic characteristic of the data assimilation system that the analysis draws nearer to the observations than the guess (RMSE and biases are smaller). This is more a basic sanity check than a real evaluation. On top of that it does not ensure that there are no spurious jumps in standard deviations and biases in the timeseries.*

[AR] Agreed. We have moved the O-A vs O-B comparison to the Supplementary Material. Additional analyses described in our reply to comment 2.1.

*2.19.     page 10, pages 3-5: this kind of information, what needs to be consolidated and presented altogether in the overview part of the document in order to get an impression what additional information is provided from the regional reanalysis (see also at specific comments)*

[AR] We assume the reviewer's comment pertains to page 10, lines 3-5. We have placed parts of this alongside the description of results. Then, an overview paragraph is added at the start of Section 3 "Preliminary evaluation", which reads
"Our evaluation focuses on three areas: surface variables, pressure-level temperature and wind, and precipitation. For the surface variables, we compare BARRA-R against point-scale observations and gridded analyses of observations for 2 m temperature. For the pressure levels, we evaluate BARRA-R against point-scale observations of temperature and wind, and examine the timeseries of the bias between BARRA-R and the global reanalyses. Finally, as rain observations are not assimilated in BARRA-R, gridded analyses of rain observations from gauges and satellites are used to provide the best independent reference in this study."

 Section 4 provides an overview summary of the additional information provided from BARRA-R.

*2.20.     page10, lines 8-9: what observations do you mean, which are not used in the analysis? Or do you mean that since it is a forecast it is an indirect relation, only?*

[AR] It is the latter. The text now reads "These observations have only an indirect relation to the forecasts as they are not used in the analysis for the associated cycle $t_0$."

*2.21.     page 11, lines 11-14: please mention "figure not shown" when you discuss a particular aspect, which is not supported by any presented figures.*

[AR] We refer to the supporting figure in the Supplementary Material.

*2.22.     page 11, lines 16-19: it would be good to include some information about the strengths and weaknesses of the AWAP dataset, which is used for validation since it influences the interpretation of the results.*

[AR] See our reply to comment 2.2.

*2.23.     page 11, lines 24-26, Figure 5: ".... BARRA-R shows cold and warm biases around 1K in daily maximum and minimum temperature, respectively..." It is quite confusing since the order of the figures is the opposite than in the text (top: minimum, bottom: maximum temperature) and the colour code in the inset is counter-intuitive (red: cold bias, blue: warm bias). I think, that the text states just the opposite than what is seen in the figure: in the minimum temperature there is a*

*cold bias (the reanalysis is colder than the AWAP dataset) and a warm bias (the reanalysis is warmer than the AWAP dataset) in the maximum temperature. Please, clarify this confusion and improve the colour scale of the Figure.*

[AR] The colour schemes in Figure 5 have been changed to use red to represent warm bias, and blue for cold bias. The result shows cold bias in TMax, and warm bias in TMin.

*2.24.    page12, lines14-19: one would expect that at least at lower levels some improvements can be detected. Maybe some discussion about this can be put here.*

[AR] We speculate that this is because we are benchmarking against ERA-Interim analysis which is already a close fit to the sonde observations. Hence the differences are not apparent at the observation locations. The text has been amended to emphasize this, it reads "There are also small differences between the analyses and BARRA-R background, indicating that the 0.11° forecast model does not degrade from the lower-resolution analysis of BARRA-R but also does not improve upon the ERA-Interim's 0.75° representation of these fields at the observation locations."

New analysis in Figure 8 shows that there are notable differences in air temperature between the reanalyses.

*2.25.    page 13, line 2: "... to adjust to remove the excess", is that correct English-wise? Please check.*

[AR] "excess" can be used as an adjective or a noun.

*2.26.    page 13, line 11: I think, there is a typo there, since it should read as "The first row...." and not the "first column".*

[AR] Agreed.

*2.27.    Figure 7: it is difficult to read these figures, maybe figures only for Australia (as in the supplementary material) would be sufficient.*

[AR] We do not wish to restrict the evaluation only on Australia. Many differences are notable in regions exterior to Australia, particularly in the tropics and New Zealand.

*2.28.    page 13, line 25: it is very difficult to see the details in Tasmania, I am not sure if such details should be discussed here.*

[AR] Zoomed in plots of Australia is provided in Supplementary Material. This is noted in the caption of Figure 9.

*2.29.    page 14, line 17: what do you mean "temperate region", please? How the latitude bands were selected (for instance 39.2 looks a bit strange choice)?*

[AR] The text has been revised to discuss the new results. The 39.2 to 49 deg South sub-domain is defined to focus on Tasmania and New Zealand.

*2.30.    page 14, lines 18-24: "grid-point storms"; as I mentioned above, I see this numerical issue as a major problem and the reanalysis should not have such numerical instability (and this should have been corrected prior to production).*

[AR] Please see our reply to comment 2.3.

*2.31.    Summary and outlook: page 15, line 2: what do you mean on "global effort"?*

[AR] The text has been simplified as "BARRA is the first regional reanalysis that focuses on the Australasian section of the Southern Hemisphere."

*2.32.    page 15, line 18: I think the wind bias should be improved with model development efforts and not via post-processing. It is true that some specific users might use postprocessing, but I think the modelling community should aim to improve the reanalysis with the improvements of the NWP methodology.*

[AR] Model development is an ongoing but very challenging endeavour, leading to different generations of NWP and reanalyses. Despite the community's best efforts however, model biases are not likely to disappear in the foreseeable future.  Yet, by working within the limitations of the models we are still able to provide valuable information to users. Therefore, practical approaches to relate model fields to observations are through post-processing and statistical methods. They are commonly used internationally by meteorological centres including the Bureau and National Weather Service, NOAA using a MOS (Model Output Statistics) method based on Glahn and Lowry (1972).

*2.33.    page 15, line 22: what is "GA6 configuration"?*

[AR] GA6 configurations are stated in section 2.1 "Forecast model".

*2.34.    page 16, line 3: I disagree that the grid-point storms should be screened out via postprocessing. This is a major numerical problem, which has to be solved before producing a reanalysis.*

[AR] Please see our reply to comment 2.3.

*2.35.    page 16, line13-14: it is important to get an overview in this paper about the relative merits between reanalysis and downscaling, since this gives justification for having reanalysis instead of simple downscaling. Therefore, some information about this issue should be provided at an early part of this paper.*

[AR] The introduction has reviewed several papers on the usefulness of regional reanalyses over dynamical downscaling, underpinning efforts around the various regional reanalysis projects internationally.

The comparisons of short-ranged O-A (observation – analysis) and O-B (observation – background) statistics in Table 1 (now moved to Table S1 of the Supplementary Material) showed that, with O-A being consistently better O-B for various observational types, an analysis within the BARRA-R system yields a more accurate short-ranged forecast than simply using the background, where the background from the previous analysis is, by extension, better than pure dynamical downscaling from the very first cycle.

*2.36.    page 16: lines 18-22: this part of the text is mostly repetition (see page 15, lines 3-7), which should be avoided.*

[AR] Agreed. The paragraph has been merged with the first paragraph of the section.

[AR] **Other significant changes not described above**
- **Figure 4**, which shows comparisons between models and observations in terms of percentiles, has been revised. An error in the codes to interpolate model grids to observing station locations has been fixed.
- **Table 1**, which lists the various observational data sources, has been revised to reflect the extension of the reanalysis period to include 2018. Data sources has also been corrected.

[revised manuscript text omitted]

**Supplementary Material**

[Figure]

**Figure S1 Spatial maps of the difference in RMSD between (left column) BARRA-R and ERA-Interim, and (right) BARRA-R and MERRA-2. RMSD is calculated on model forecasts against observed (a) 2 m temperature, (b) 10 m wind speed and (c) surface pressure. These figures supplementing Figure 3.**

[Figure]

**Figure S2 As Figure 8 but over Australia only. (i) Mean annual precipitation [mm], and (ii) fractions of rain days, (iii) heavy precipitation days and (iv) very heavy precipitation days, from 2007 to 2016 from (a) AWAP, (b) GPCC, (c) TMPA, (d) ERA-Interim, and (e) BARRA-R. Regions with more than 10% missing values in AWAP are masked.**

[Figure]

**Figure S3 Mean difference in daily precipitation between BARRA-R and TMPA from 2007-2016 over two sub-domains in the tropics and New Zealand. The data is regridded to the TMPA 0.25° grid.**

| Fields | O-B | | O-A | |
|---|---|---|---|---|
| | **Bias** | **RMSD** | **Bias** | **RMSD** |
| Surface temperature [K] | -0.09 | 1.78 | -0.10 | **1.61** |
| Surface pressure [Pa] | -3.67 | 101.69 | **-2.08** | **68.85** |
| Surface relative humidity [%] | 0.0 | 10.0 | 0.00 | **8.0** |
| Surface zonal wind [m/s] | 0.05 | 1.97 | **-0.01** | **1.74** |
| Surface meridional wind [m/s] | 0.04 | 1.94 | **0.01** | **1.72** |
| Aircraft potential temperature [K] | -0.24 | 1.34 | **-0.17** | **1.10** |
| Aircraft zonal wind [m/s] | -0.04 | 3.05 | **-0.03** | **2.09** |
| Aircraft meridional wind [m/s] | -0.18 | 3.06 | **-0.07** | **2.07** |
| Sonde temperature at 980 hPa [K] | -0.15 | 1.11 | **-0.08** | **0.81** |
| Sonde temperature at 500 hPa [K] | -0.33 | 0.92 | **-0.18** | **0.60** |
| Sonde zonal wind at 980 hPa [m/s] | -0.15 | 2.45 | **-0.06** | **1.45** |
| Sonde zonal wind at 500 hPa [m/s] | -0.17 | 2.52 | **-0.07** | **1.41** |
| Sonde meridional wind at 980 hPa [m/s] | 0.23 | 2.34 | **0.09** | **1.38** |
| Sonde meridional wind at 500 hPa [m/s] | 0.11 | 2.44 | **0.03** | **1.39** |
| Satwind zonal wind [m/s] | 0.36 | 3.16 | **0.27** | **2.72** |
| Satwind meridional wind [m/s] | 0.05 | 2.90 | **0.01** | **2.40** |
| Scatwind zonal wind [m/s] | 0.06 | 1.39 | **0.03** | **0.95** |
| Scatwind meridional wind [m/s] | 0.20 | 1.78 | **-0.02** | **1.32** |

**Table S1 Comparisons of the 10-year (2007-2016) mean of the RMSD and bias between the analyses and observations (O-A) and those between the background and observations (O-B), calculated for various observational types across the BARRA-R domain. Bold values show reduction in the RMSD and the magnitude of the bias by the analyses, i.e., the analyses draw the model forecasts closer to these observation types.**

---

## Author Comment (AC3) · 18 Mar 2019

**Response to Reviewer #3**

**1. Response to general comment**
*The paper is well written and seems to contain almost no typos or errors in its discussion and references. It describes a new extensive data set which is very valuable for the region in question and is novel in this way. The results are quite well described and documented. The evaluation methodologies are well thought through and made in a careful way. I have only some comments and questions as below:*

[AR] We thank the reviewer for their positive feedback on our manuscript. We have made changes to the manuscript to improve its clarity.

**2. Response to specific comments**

*2.1. Spin-up period of one month seems short comparing other reanalyses and the reference to Renshaw 2013 may not be so valid today? It would be worthwhile to revisit this point if at all possible, by allowing longer spin up or comparing the first year/season with a run that has had several years. The question is mainly about soil properties which at least at higher latitudes have a long time constant.*

[AR] The land state and in particular sub surface soil moisture can have a very long memory. One month is indeed likely to be too short to adequately spin-up the BARRA soil moisture. To ameliorate this problem, BARRA soil moisture was initialised using soil moisture analyses from an off-line JULES simulation. This off-line JULES simulation used a 10-year long spin-up period and then was run continuously for the 1990 to 2014 period. Even so, we recognise that BARRA will still need to spin up from the offline/external soil moisture data, which is finer in resolution than ERA-Interim but remains > 12 km. While one-month of runs is sufficient to allow the soil moisture in the top layers to stabilize, it is insufficient for bottom layers. We expanded our discussion in Section 2.1.2 "Soil moisture":

"For the 1990 to 2014 period, soil moisture fields in BARRA-R are initialised daily at every 06 UTC using soil moisture analyses from an offline simulation of JULES, at 60 km resolution, driven by bias corrected ERA-Interim atmosphere forcing data, using methods described in Dharssi and Vinodkumar (2017) and Zhao et al. (2017). The simulation used a 10-year long spin-up period and then was run continuously for the 1990 to 2014 period. The near-surface soil moisture analyses are found to have good skill for the Australian region when validated against ground-based soil moisture observations (Dharssi and Vinodkumar, 2017). As the offline runs were terminated at the end of December 2014, the daily initialization scheme is continued with soil moisture analyses from the Bureau's global NWP system – ACCESS-G (Bureau of Meteorology, 2016). These external soil moisture analyses are downscaled to the BARRA-R grid using a simple method that takes into account differences in soil texture. As well, in each 6-hourly cycle, a land surface analysis is conducted within BARRA (see Section 2.2). The daily initialisation was conducted with the purpose of avoiding spurious drift in the BARRA moisture fields and reducing the time needed to spin up from ERA-Interim initial conditions. However, as multiple parallel production streams are needed to produce the reanalysis (see Section 2.2), a discontinuity in soil moisture in the bottom two layers exists between successive production streams, although soil moisture in the top two layers become stable after one-month of runs. A discontinuity occurring at the 2014-2015 changeover has recently been reported by BARRA data users. These impacts, particularly on forested regions where trees extract water from the deep soil layers, are under investigation."

*2.2. Page 12, line 27: will be erroneous→will be more erroneous*

[AR] Agreed.

*2.3. Page 13, line 29: TMPA over land where the rain gauge data .... : does TMPA use rain gauge data ? Probably but you describe as multi-satellite analysis; please clarify.*

[AR] TMPA's 3B42 multi-satellite product uses monthly bias correction with rain gauge observations. The added text reads "The TMPA 3B42 combines precipitation estimates from various satellite systems and land surface rain gauge monthly analysis."

[AR] **Other significant changes not described above**
- Section 3.1.2 "Comparisons with gridded analysis of observed 2 m temperature has been extended to look at the 14-year changes to biases in TMax (daily maximum temperature) and TMin (daily minimum temperature) relative to AWAP – in new **Figure 6**. We found shifts in biases in BARRA-R, but also in the global reanalyses ERA-Interim and MERRA-2. Further, these changes do not coincide with our changes in soil moisture initialization in 2014-2015 or OSTIA data.
- Section 3.2 "Pressure levels" has been extended to look at 14-year changes to air temperature at 500, 700 and 850 hPa pressure levels – new **Figure 8**. We found shifts in biases between BARRA-R and ERA-Interim from 2010, but these shifts are absent when comparing BARRA-R with MERRA-2. The results are therefore inconclusive.
- Section 3.3 "Precipitation" has been extended to look at 14-year differences in monthly precipitation between BARRA-R and TMPA, ERA-Interim and TMPA, BARRA-R and GPCC, and ERA-Interim and GPCC – revised **Figure 10 and 11**. The most apparent shifts in precipitation occur in north Australia, where BARRA-R shows wetter summer events after 2010, at the end of the Australian millennium drought and onset of La Nina.
- **Figure 4**, which shows comparisons between models and observations in terms of percentiles, has been revised. An error in the codes to interpolate model grids to observing station locations has been fixed.
- **Table 1**, which lists the various observational data sources, has been revised to reflect the extension of the reanalysis period to include 2018. Data sources has also been corrected.

[revised manuscript text omitted]

**Supplementary Material**

[Figure]

**Figure S1 Spatial maps of the difference in RMSD between (left column) BARRA-R and ERA-Interim, and (right) BARRA-R and MERRA-2. RMSD is calculated on model forecasts against observed (a) 2 m temperature, (b) 10 m wind speed and (c) surface pressure. These figures supplementing Figure 3.**

[Figure]

**Figure S2 As Figure 8 but over Australia only. (i) Mean annual precipitation [mm], and (ii) fractions of rain days, (iii) heavy precipitation days and (iv) very heavy precipitation days, from 2007 to 2016 from (a) AWAP, (b) GPCC, (c) TMPA, (d) ERA-Interim, and (e) BARRA-R. Regions with more than 10% missing values in AWAP are masked.**

[Figure]

**Figure S3 Mean difference in daily precipitation between BARRA-R and TMPA from 2007-2016 over two sub-domains in the tropics and New Zealand. The data is regridded to the TMPA 0.25° grid.**

| Fields | O-B | | O-A | |
| --- | --- | --- | --- | --- |
| | **Bias** | **RMSD** | **Bias** | **RMSD** |
| Surface temperature [K] | -0.09 | 1.78 | -0.10 | **1.61** |
| Surface pressure [Pa] | -3.67 | 101.69 | **-2.08** | **68.85** |
| Surface relative humidity [%] | 0.0 | 10.0 | 0.00 | **8.0** |
| Surface zonal wind [m/s] | 0.05 | 1.97 | **-0.01** | **1.74** |
| Surface meridional wind [m/s] | 0.04 | 1.94 | **0.01** | **1.72** |
| Aircraft potential temperature [K] | -0.24 | 1.34 | **-0.17** | **1.10** |
| Aircraft zonal wind [m/s] | -0.04 | 3.05 | **-0.03** | **2.09** |
| Aircraft meridional wind [m/s] | -0.18 | 3.06 | **-0.07** | **2.07** |
| Sonde temperature at 980 hPa [K] | -0.15 | 1.11 | **-0.08** | **0.81** |
| Sonde temperature at 500 hPa [K] | -0.33 | 0.92 | **-0.18** | **0.60** |
| Sonde zonal wind at 980 hPa [m/s] | -0.15 | 2.45 | **-0.06** | **1.45** |
| Sonde zonal wind at 500 hPa [m/s] | -0.17 | 2.52 | **-0.07** | **1.41** |
| Sonde meridional wind at 980 hPa [m/s] | 0.23 | 2.34 | **0.09** | **1.38** |
| Sonde meridional wind at 500 hPa [m/s] | 0.11 | 2.44 | **0.03** | **1.39** |
| Satwind zonal wind [m/s] | 0.36 | 3.16 | **0.27** | **2.72** |
| Satwind meridional wind [m/s] | 0.05 | 2.90 | **0.01** | **2.40** |
| Scatwind zonal wind [m/s] | 0.06 | 1.39 | **0.03** | **0.95** |
| Scatwind meridional wind [m/s] | 0.20 | 1.78 | **-0.02** | **1.32** |

**Table S1 Comparisons of the 10-year (2007-2016) mean of the RMSD and bias between the analyses and observations (O-A) and those between the background and observations (O-B), calculated for various observational types across the BARRA-R domain. Bold values show reduction in the RMSD and the magnitude of the bias by the analyses, i.e., the analyses draw the model forecasts closer to these observation types.**

---

## Author Comment (AC4) · 18 Mar 2019

**Response to Reviewer #4**

**1. Response to general comment**
*This paper describes the development of the BARRA system for producing atmospheric reanalyses over the Australasian region. The manuscript is well written and the topic is of interest to the atmospheric modelling community. However, I believe the manuscript could be slightly improved with some minor changes.*

[AR] We thank the reviewer for their positive feedback on our manuscript. We use this opportunity to improve the clarity of the manuscript.

**2. Response to specific comments**
*2.1. (p 3) The authors refer to a seamless model system in the introduction. Given that there are obvious differences between how the reanalysis, NWP and (presumably) regional climate modelling systems are configured, could the authors explain more clearly the advantages of a seamless modelling system?*

[AR] There are many potential advantages of using a seamless approach to weather and climate prediction. These are described in Brown et al. (2012). One advantage is that by using the same modelling system in both areas, it can aid in understanding the variability of error across multiple space and time scales. We revised the text in the introduction to make this clearer. It now reads " At the same time, embedded forecast models can be used within the framework of the Coordinated Regional Climate Downscaling Experiment (CORDEX) (Martynov et al., 2013) within a seamless framework for weather and climate prediction, where model deficiency in the individual areas that differ in spatial and time scales, can be more readily understood (Brown et al., 2012)."

*2.2. (section 2) Could the authors provide a short description on the computational requirements of the BARRA system? For example, could the authors comment on how many cores where used and at what rate can reanalysis data be generated?*

[AR] We decided not to include this information in the manuscript itself, because the paper focuses on the science contributions. The computational requirements of BARRA-R depend on specific hardware and software (including compilers and level of optimisation), which we think are less important for most readers of this manuscript – data users. We provide a description here: BARRA-R is produced on Intel Xeon Broadwell (2.6 GHz) based nodes, each with 28 cores, at the National Computational Infrastructure (NCI, www.nci.org.au). Each analysis uses about 700 CPUs for around 30 minutes, and each forecast run uses about 900 CPUs for 20 minutes. We perform well over 42,000 cycles (of analysis and forecast runs) to complete 29-year BARRA-R reanalysis. The throughput, ranging from 20-50 reanalysis days per day, had varied across the production period due to highly variable loads on the HPC, and changes in production setup (i.e., number of parallel production streams).

[AR] **Other significant changes not described above**
- Section 3.1.2 "Comparisons with gridded analysis of observed 2 m temperature has been extended to look at the 14-year changes to biases in TMax (daily maximum temperature) and TMin (daily minimum temperature) relative to AWAP – in new **Figure 6**. We found shifts in biases in BARRA-R, but also in the global reanalyses ERA-Interim and MERRA-2. Further, these changes do not coincide with our changes in soil moisture initialization in 2014-2015 or OSTIA data.
- Section 3.2 "Pressure levels" has been extended to look at 14-year changes to air temperature at 500, 700 and 850 hPa pressure levels – new **Figure 8**. We found shifts in

biases between BARRA-R and ERA-Interim from 2010, but these shifts are absent when comparing BARRA-R with MERRA-2. The results are therefore inconclusive.

- Section 3.3 "Precipitation" has been extended to look at 14-year differences in monthly precipitation between BARRA-R and TMPA, ERA-Interim and TMPA, BARRA-R and GPCC, and ERA-Interim and GPCC – revised **Figure 10 and 11**. The most apparent shifts in precipitation occur in north Australia, where BARRA-R shows wetter summer events after 2010, at the end of the Australian millennium drought and onset of La Nina.
- **Figure 4**, which shows comparisons between models and observations in terms of percentiles, has been revised. An error in the codes to interpolate model grids to observing station locations has been fixed.
- **Table 1**, which lists the various observational data sources, has been revised to reflect the extension of the reanalysis period to include 2018. Data sources has also been corrected.

[revised manuscript text omitted]

**Supplementary Material**

[Figure]

Figure S1 Spatial maps of the difference in RMSD between (left column) BARRA-R and ERA-Interim, and (right) BARRA-R and MERRA-2. RMSD is calculated on model forecasts against observed (a) 2 m temperature, (b) 10 m wind speed and (c) surface pressure. These figures supplementing Figure 3.

[Figure]

**Figure S2 As Figure 8 but over Australia only. (i) Mean annual precipitation [mm], and (ii) fractions of rain days, (iii) heavy precipitation days and (iv) very heavy precipitation days, from 2007 to 2016 from (a) AWAP, (b) GPCC, (c) TMPA, (d) ERA-Interim, and (e) BARRA-R. Regions with more than 10% missing values in AWAP are masked.**

[Figure]

**Figure S3 Mean difference in daily precipitation between BARRA-R and TMPA from 2007-2016 over two sub-domains in the tropics and New Zealand. The data is regridded to the TMPA 0.25° grid.**

| Fields | O-B | | O-A | |
|---|---|---|---|---|
| | Bias | RMSD | Bias | RMSD |
| Surface temperature [K] | -0.09 | 1.78 | -0.10 | **1.61** |
| Surface pressure [Pa] | -3.67 | 101.69 | **-2.08** | **68.85** |
| Surface relative humidity [%] | 0.0 | 10.0 | 0.00 | **8.0** |
| Surface zonal wind [m/s] | 0.05 | 1.97 | **-0.01** | **1.74** |
| Surface meridional wind [m/s] | 0.04 | 1.94 | **0.01** | **1.72** |
| Aircraft potential temperature [K] | -0.24 | 1.34 | **-0.17** | **1.10** |
| Aircraft zonal wind [m/s] | -0.04 | 3.05 | **-0.03** | **2.09** |
| Aircraft meridional wind [m/s] | -0.18 | 3.06 | **-0.07** | **2.07** |
| Sonde temperature at 980 hPa [K] | -0.15 | 1.11 | **-0.08** | **0.81** |
| Sonde temperature at 500 hPa [K] | -0.33 | 0.92 | **-0.18** | **0.60** |
| Sonde zonal wind at 980 hPa [m/s] | -0.15 | 2.45 | **-0.06** | **1.45** |
| Sonde zonal wind at 500 hPa [m/s] | -0.17 | 2.52 | **-0.07** | **1.41** |
| Sonde meridional wind at 980 hPa [m/s] | 0.23 | 2.34 | **0.09** | **1.38** |
| Sonde meridional wind at 500 hPa [m/s] | 0.11 | 2.44 | **0.03** | **1.39** |
| Satwind zonal wind [m/s] | 0.36 | 3.16 | **0.27** | **2.72** |
| Satwind meridional wind [m/s] | 0.05 | 2.90 | **0.01** | **2.40** |
| Scatwind zonal wind [m/s] | 0.06 | 1.39 | **0.03** | **0.95** |
| Scatwind meridional wind [m/s] | 0.20 | 1.78 | **-0.02** | **1.32** |

**Table S1 Comparisons of the 10-year (2007-2016) mean of the RMSD and bias between the analyses and observations (O-A) and those between the background and observations (O-B), calculated for various observational types across the BARRA-R domain. Bold values show reduction in the RMSD and the magnitude of the bias by the analyses, i.e., the analyses draw the model forecasts closer to these observation types.**

---

## Referee Report (RR1)

*The main question for a regional reanalysis is to clearly demonstrate whether the use of such system is justified, which means that more value can be added to the global reanalysis then it would be the case with a pure dynamical downscaling. For this question one has to understand the additional information brought into the regional system in terms of more precise dynamical and physical description of the atmosphere, but also in terms of additional and advanced use of observations. I miss a summary of this kind from the manuscript though some of these aspects are highlighted here and there in the paper.*

[AR] The introduction has reviewed several papers on the usefulness of regional reanalyses over dynamical downscaling, underpinning efforts around the various regional reanalysis projects internationally.

*My point here was not a general assessment of the value of regional reanalysis with respect to dynamical downscaling, but a particular one which analyses the merit of BARRA in that regard.*

The comparisons of short-ranged O-A (observation – analysis) and O-B (observation – background) statistics in Table 1 (now moved to Table S1 of the Supplementary Material) showed that, with O-A being consistently better O-B for various observational types, an analysis within the BARRA-R system yields a more accurate short-ranged forecast than simply using the background, where the background from the previous analysis is, by extension, better than pure dynamical downscaling from the very first cycle.

*I think, the fact that O-A is better than O-B does not show that the reanalysis is better than dynamical downscaling (it was also admitted by the authors answering to another question of my original review). In case of dynamical downscaling there is a higher resolution dynamics and physics of the model and the surface characteristics are described in more details (I mean on dynamical downscaling that a model is used to downscale the lower resolution information possibly also taking into account a better surface description).*

*2.3. The existence of the "grid-point storms" is embarrassing since such numerical problems should not happen in a reanalysis, where a robust and properly (thoroughly) tested NWP system should be used. Normally, the reanalysis should not be run if such problems are not yet solved. There is a need for a thorough explanation how this could happen and how this deficiency compromises the validity of the reanalysis results.*

[AR] The Unified Model is sufficiently robust to be useful for many operational meteorological centres in Australia, UK, India, Singapore, Korea, South Africa and New Zealand. The issue of "grid- point storms" is also not unique to UM but for instance, also occurs in the widely-used Weather Research and Forecasting (WRF) model from NCAR. When the convective (sub-grid) parameterization scheme in non-convective resolving models does not stabilize the air column, meteorological events can develop at the smallest resolvable scales in the model, producing unrealistically strong vertical velocities and precipitation (Scinocca and McFarlane, 2004; Williamson, 2013). The resulting "grid-point storms" occur more readily in models with higher horizontal resolutions (Williamson, 2013). The issue becomes unavoidable for BARRA-R as it aims to be sufficiently higher resolution than global reanalyses but could not be sufficiently high resolution (< 2 km) (and computationally prohibitive) to resolve convection explicitly without the need for a convective parameterization scheme.

Further, we do not think that the wet biases in BARRA-R over the tropics and New Zealand are entirely due to grid point storms. Additional analyses have been made to identify the location of

precipitation excess in the tropics and New Zealand (Figure S3 in Supplementary Material). We found that the higher precipitation in BARRA-R are concentrated at high or sharp topographical regions in PNG, Indonesia, Sumatra and small Indonesian Islands, and west coast and Southern Alps of New Zealand. At these locations, GPCC (gauge analysis) and TMPA would underestimate the precipitation. With these considerations, the actual levels of bias observed in BARRA-R are not entirely clear.

*I am still NOT convinced at all that the grid-point storms are unavoidable details of a numerical model. These are really numerical artefacts, which should be avoided! Regarding the answer of the authors:*

- *It is not an argument that other models (e.g. WRF) and other centres (UK, India, Singapore, Korea, South Africa, New Zealand) have the same problem. This is not an answer to the question!*
- *As the authors properly mention this problem is coming from the discrepancy between the convection scheme and the non-convective resolving model. It is well-known that in the so called grey resolution zone (typically around 3-7km resolution range) adequate convection scheme should be used. The occurrence of the grid-point storms indicate that the applied convection scheme is not suited to that resolution!*

*I think the only way to circumvent this issue in the article is (i) admit this problem (which is already the case in the manuscript), (ii) properly explain its origin, (iii) warn the users particularly if they would like to have a local evaluation and (iv) convince the readers/users that this problem does not have a significant impact on the climate quality of the reanalysis. But, please don't use such arguments that it is also apparent in other models and centres!!*

*2.8. page 2, line 25: please give reference for the Copernicus reanalysis*

[AR] Agreed. I have added a reference to Ridal et al. (2017).

*Ridal et al (2017) is a reference to UERRA and not to Copernicus reanalysis (ERA5). Use for instance Hersbach and Dee (2016).*

2.35. *page 16, line13-14: it is important to get an overview in this paper about the relative merits between reanalysis and downscaling, since this gives justification for having reanalysis instead of simple downscaling. Therefore, some information about this issue should be provided at an early part of this paper.*

[AR] The introduction has reviewed several papers on the usefulness of regional reanalyses over dynamical downscaling, underpinning efforts around the various regional reanalysis projects internationally.

*Again, I mean this particularly for BARRA and not in general!*

The comparisons of short-ranged O-A (observation – analysis) and O-B (observation – background) statistics in Table 1 (now moved to Table S1 of the Supplementary Material) showed that, with O-A being consistently better O-B for various observational types, an analysis within the BARRA-R system yields a more accurate short-ranged forecast than simply using the background, where the background from the previous analysis is, by extension, better than pure dynamical downscaling from the very first cycle.

*See my feedback above for the same issue!*

*Two small additional issues: please use ERA5 without hyphen and I think one has to use short-range instead of short-ranged.*

---

## Author Response (AR2)

Response to Reviewer #2 (Second round)

[RC1] indicates the reviewer comments from the first round
[RC2] indicates the reviewer comments from the second round
[AR1] indicates authors' reply to the first round
[AR2] indicates authors' reply to the second round

[RC2] EVALUATION SUMMARY: I would like to thank to the authors for their hard work to update their paper. Generally, I am happy with the answers to my questions, but I think there are still two issues, which needs some more attention:
1. There is no particular analysis why BARRA is expected to provide additional information with respect to its global counterparts and with respect to dynamical downscaling. Only generalities are mentioned here and I would be interested in the particular aspects of BARRA at that regard.
2. The answer for the grid-point storm question is not satisfactory in my opinion. Please don't try to convince the readers that it is normal in an NWP model to have grid-point storms. This is a numerical problem, which must be avoided. It is coming from the fact that the convection scheme (and the parameterisation schemes in general) is not suited to the resolution used for the reanalysis. Normally the convection scheme should have adapted to have the proper match!

[AR2] We appreciate the reviewer's time to provide their comments in this second round. Please see below for our specific replies.

[RC2] So overall, I am still not fully happy with the paper, but due to the very positive attitude of the other three reviewers I don't want to block the paper from publication. Therefore, I suggest, to accept the paper provided a proper thought is given to these two issues. See below my original comments (in italics) the answer of the authors and my latest feedback (red italics).
The main question for a regional reanalysis is to clearly demonstrate whether the use of such system is justified, which means that more value can be added to the global reanalysis then it would be the case with a pure dynamical downscaling. For this question one has to understand the additional information brought into the regional system in terms of more precise dynamical and physical description of the atmosphere, but also in terms of additional and advanced use of observations. I miss a summary of this kind from the manuscript though some of these aspects are highlighted here and there in the paper.
[AR1] The introduction has reviewed several papers on the usefulness of regional reanalyses over dynamical downscaling, underpinning efforts around the various regional reanalysis projects internationally.
[RC1] My point here was not a general assessment of the value of regional reanalysis with respect to dynamical downscaling, but a particular one which analyses the merit of BARRA in that regard.
[AR1] The comparisons of short-ranged O-A (observation – analysis) and O-B (observation – background) statistics in Table 1 (now moved to Table S1 of the Supplementary Material) showed that, with O-A being consistently better O-B for various observational types, an analysis within the BARRA-R system yields a more accurate short-ranged forecast than simply using the background, where the background from the previous analysis is, by extension, better than pure dynamical downscaling from the very first cycle.
[RC2] I think, the fact that O-A is better than O-B does not show that the reanalysis is better than dynamical downscaling (it was also admitted by the authors answering to another question of my original review). In case of dynamical downscaling there is a higher resolution dynamics and physics of the model and the surface characteristics are described in more details (I mean on dynamical downscaling that a model is used to downscale the lower resolution information possibly also taking into account a better surface description).

[AR2] The purpose of this paper is to document an existing product to assist in assuring that it used appropriately to assist further scientific inquiry and decision making by a variety of users. As such, we have endeavoured to establish BARRA as a credible reanalysis that adds value to existing and widely used data sets (i.e., ERA-Interim and MERRA-2), and that this was done with a system that reflects world best practice in analysing and modelling the atmosphere, namely the Unified Modelling (UM) System.

Within the confines of the UM System, the approach in BARRA-R has been agreed as the most appropriate method within the Unified Modelling Partnership (i.e. the consolidated experts on using the Unified Modelling System), as shown by the use both as a candidate for UERRA and for the IMDAA reanalysis. We are however not proposing that this is the best or most efficient solution, but rather a useful and affordable solution. Comparisons with other methods would require the involvement of other experts in the use of these modelling systems, under carefully controlled conditions regarding common data sets etc. Such an intercomparison is beyond the scope of this work.

Based on RC2, the reviewer may also be mistaken. The higher resolution (12km) modelling system is used throughout the warm running analysis-forecast system that comprises the reanalysis, so the high resolution dynamics, model physics and surface characterization are all inherent to the entire reanalysis process. That is, the background forecast (B) and analysis (A) used in calculating the O-B and O-A statistics are based on same model and resolution. It is also outside the scope of this work to compare 12 km BARRA-R reanalysis against dynamical downscaling methods at finer resolution < 12 km.

We have added a sentence in Section 2.2 (Data assimilation system) to refer to the O-A/O-B results in the Supplementary Material. It reads, "*The observation departure statistics of the analysis, which are differences between the analysis and observations, are shown to be less than those of the model background in the Supplementary Material (Table S1). The assimilation is therefore behaving as desired by drawing the model towards observations for nearly all observational types.*"

Further, the caption of Table S1 reads "*Table S1 Comparisons of the 10-year (2007-2016) mean of the RMSD and bias between the analyses and observations (O-A) and those between the background and observations (O-B), calculated for various observational types across the BARRA-R domain. Bold values show reduction in the RMSD and the magnitude of the bias by the analyses, i.e., the analyses draw the model forecasts closer to these observation types.*"

[RC1] 2.3. The existence of the "grid-point storms" is embarrassing since such numerical problems should not happen in a reanalysis, where a robust and properly (thoroughly) tested NWP system should be used. Normally, the reanalysis should not be run if such problems are not yet solved. There is a need for a thorough explanation how this could happen and how this deficiency compromises the validity of the reanalysis results.

[AR1] The Unified Model is sufficiently robust to be useful for many operational meteorological centres in Australia, UK, India, Singapore, Korea, South Africa and New Zealand. The issue of "grid-point storms" is also not unique to UM but for instance, also occurs in the widely-used Weather Research and Forecasting (WRF) model from NCAR. When the convective (sub-grid) parameterization scheme in non-convective resolving models does not stabilize the air column, meteorological events can develop at the smallest resolvable scales in the model, producing unrealistically strong vertical velocities and precipitation (Scinocca and McFarlane, 2004; Williamson, 2013). The resulting "grid-point storms" occur more readily in models with higher horizontal resolutions (Williamson, 2013). The issue becomes unavoidable for BARRA-R as it aims to be sufficiently higher resolution than global reanalyses but could not be sufficiently high resolution (< 2

km) (and computationally prohibitive) to resolve convection explicitly without the need for a convective parameterization scheme.

Further, we do not think that the wet biases in BARRA-R over the tropics and New Zealand are entirely due to grid point storms. Additional analyses have been made to identify the location of precipitation excess in the tropics and New Zealand (Figure S3 in Supplementary Material). We found that the higher precipitation in BARRA-R are concentrated at high or sharp topographical regions in PNG, Indonesia, Sumatra and small Indonesian Islands, and west coast and Southern Alps of New Zealand. At these locations, GPCC (gauge analysis) and TMPA would underestimate the precipitation. With these considerations, the actual levels of bias observed in BARRA-R are not entirely clear.

[RC2] I am still NOT convinced at all that the grid-point storms are unavoidable details of a numerical model. These are really numerical artefacts, which should be avoided! Regarding the answer of the authors:

It is not an argument that other models (e.g. WRF) and other centres (UK, India, Singapore, Korea, South Africa, New Zealand) have the same problem. This is not an answer to the question!

As the authors properly mention this problem is coming from the discrepancy between the convection scheme and the non-convective resolving model. It is well-known that in the so called grey resolution zone (typically around 3-7km resolution range) adequate convection scheme should be used. The occurrence of the grid-point storms indicate that the applied convection scheme is not suited to that resolution!

I think the only way to circumvent this issue in the article is (i) admit this problem (which is already the case in the manuscript), (ii) properly explain its origin, (iii) warn the users particularly if they would like to have a local evaluation and (iv) convince the readers/users that this problem does not have a significant impact on the climate quality of the reanalysis. But, please don't use such arguments that it is also apparent in other models and centres!!

[AR2] We explained in AR1 that grid point storms have been documented for UM and WRF models. Met centres continue to use them in operations in spite of this, because the models have proven useful in many aspects. While a more diffusive scheme can be used to avoid this issue completely, this degrades other aspects of the model. Our explanation is to address the reviewer's perception (from [RC1]: The existence of the "grid-point storms" is embarrassing since such numerical problems should not happen in a reanalysis, where a robust and properly (thoroughly) tested NWP system should be used.). To some people, "grid point storm" is interpreted to occur at model crash due to local excessively strong convection; in which case, the model is not considered robust. In our case, we use the term to refer to spotty excessive rainfall events that do not cause model crashes. The mismatch between our and reviewer's expectations may also stem from this difference in interpretation of what grid-point storms mean.

Further, the use of UM's convection parametrisation scheme and the 12 km resolution of the model configuration are typical, performed in many prior works, including Chan et al. (2004), Mahood et al. (2018, IMDAA reanalysis), and Jermey and Renshaw (2016, UKMO reanalysis in UERRA).

(i) We have already noted the occurrence of grid point storms, which can explain part of the biases in BARRA-R, relative to observational data sets.

(ii) In the revised manuscript, we have explained the origin of this model artefacts using the added text "*When the convective parameterization in non-convective resolving models does not stabilize the air column, meteorological events can develop at the smallest resolvable scales in the model, producing unrealistically strong vertical velocities and precipitation; this is known as "grid-point storms" (Scinocca and McFarlane, 2004; Williamson, 2013; Chan et al., 2014). In our cases, the model only produces isolated excessively intense rainfall over the steep topography. Such storms occur more*

*readily in models with higher horizontal resolutions (Williamson, 2013). As the resolution increases, resolved motions can produce moisture convergence and increase CAPE very rapidly, and the rate at which column instability is produced depends on the scale of moisture and heat convergence. This also tends to occur over tropical land areas, over steep topography, and during the warm seasons, when the atmosphere is unstable and there is sufficient warm moisture supply at the surface.*

In section 2.1 (Forecast model), we added "*Our choice of the horizontal resolution follows the deterministic component of the UKMO reanalysis and the IMDAA reanalyses.*"

(iii) We have raised a few aspects (including grid point storms) in BARRA-R that differ from other reanalyses and observational data sets. As with any model data, local evaluation should be conducted before using them. In the conclusion of the revised manuscript, we added "*Given all the above considerations, local evaluation of BARRA-R reanalysis before application is recommended.*"

(iv) We have shown that the grid-point storms possibly have impacts on the rainfall averages (Figure 11). Given the rainfall amounts, they can affect studies of rainfall extremes. We now added to the conclusion sentences that read "*The characteristics of grid-point storms in terms of superficial spatial localization, precipitation amount and vertical wind speed, could be detected and screened out via post-processing. It is important as this model artefact affects the analyses of the rainfall averages and extremes.*" It is however beyond the scope of this work to conduct further evaluation.

[RC1] 2.8. page 2, line 25: please give reference for the Copernicus reanalysis
[AR1] Agreed. I have added a reference to Ridal et al. (2017).
[RC2] Ridal et al (2017) is a reference to UERRA and not to Copernicus reanalysis (ERA5). Use for instance Hersbach and Dee (2016).
[AR2] The reviewer is mistaken. The text at page 2, line 25 in the original manuscript describes UERRA, not ERA5.

The reference Hersbach and Dee (2016) is cited in the Introduction where ERA5 is noted.

[RC1] 2.35. page 16, line13-14: it is important to get an overview in this paper about the relative merits between reanalysis and downscaling, since this gives justification for having reanalysis instead of simple downscaling. Therefore, some information about this issue should be provided at an early part of this paper.
[AR1] The introduction has reviewed several papers on the usefulness of regional reanalyses over dynamical downscaling, underpinning efforts around the various regional reanalysis projects internationally.
[RC2] Again, I mean this particularly for BARRA and not in general!
[AR1] The comparisons of short-ranged O-A (observation – analysis) and O-B (observation – background) statistics in Table 1 (now moved to Table S1 of the Supplementary Material) showed that, with O-A being consistently better O-B for various observational types, an analysis within the BARRA-R system yields a more accurate short-ranged forecast than simply using the background, where the background from the previous analysis is, by extension, better than pure dynamical downscaling from the very first cycle.
[RC2] See my feedback above for the same issue!
[AR2] See our reply above.

[RC2] Two small additional issues: please use ERA5 without hyphen and I think one has to use short-range instead of short-ranged.
[AR2] These have been corrected.

[revised manuscript text omitted]